# Efficiently Estimating Data Efficiency for Language Model Fine-tuning

## Abstract

While large language models (LLMs) demonstrate reasonable zero-shot capability across many downstream tasks, fine-tuning is a common practice to improve their performance. However, a task's *data efficiency* — i.e., the number of fine-tuning examples needed to achieve a desired level of performance — is often unknown, resulting in costly cycles of incremental annotation and retraining. Indeed, we demonstrate across a curated set of 30 specialized tasks that performant LLMs may struggle zero-shot but can attain stronger performance after fine-tuning. This motivates the need for methods to predict a task's data efficiency *without* requiring incremental annotation. After introducing a concrete metric that quantifies a task's data efficiency, we propose using the *gradient cosine similarity of low-confidence examples* as a way to predict data efficiency based on a small number of labeled samples. We validate our approach on the collected set of tasks with varying data efficiencies, attaining 8.6% error in overall data efficiency prediction and eliminating hundreds of unnecessary annotations. Our experiment results and implementation code are available in the supplementary material.

## 1 Introduction

Large language models (LLMs) are increasingly treated as generalist systems that can competently perform any text-based task zero-shot, i.e., without requiring any task-specific training data (Brown et al., 2020). However, the zero-shot performance of an LLM often lags behind human-level (or otherwise acceptable) performance (Li et al. (2023); Liu et al. (2022b); Ouyang et al. (2022); Sanh et al. (2022); Singhal et al. (2023); Wei et al. (2022)). In such cases, fine-tuning on task-specific data can provide a simple way to improve an LLM's performance by reinforcing the specified format of the model response (Ouyang et al. (2022); Sanh et al. (2022); Wei et al. (2022)) or specializing the LLM to the task (Li et al. (2023); Liu et al. (2022b); Singhal et al. (2023)). Indeed, fine-tuning a pre-trained LLM can require orders of magnitude less task-specific data than training on the task from scratch. Zhou et al. (2023) show that an LLM can easily learn to output high-quality responses with only hundreds or thousands of examples, which Aghajanyan et al. (2020) suggests is enabled by the pretraining phase compressing large-scale knowledge and reducing the downstream task's intrinsic dimensionality.

A key consideration when fine-tuning LLMs is the task's *data efficiency*, i.e., the number of task-specific labeled data points required to reach a desired performance level. Unfortunately, the data efficiency of a given task is generally not clear a priori – as we show in Section 2, some tasks require only a few dozens of samples to reach or exceed human-level performance while others may require thousands. A straightforward way of determining a task's data efficiency is to collect a large pool of labeled data and fine-tune the model at various data budgets, evaluating performance at each budget and determining the amount of data required to reach desired performance. However, this approach requires annotating a large training dataset and fine-tuning many models, obviating the purpose of estimating the data efficiency in the first place. Fine-tuning scaling laws can be fit to explore the relationship between the model loss and fine-tuning data size (Zhang et al., 2024), but fitting these scaling laws involves specific parameters unknown before fine-tuning on the downstream task. We argue that a useful method for predicting fine-tuning data efficiency should be able to do so *efficiently* – i.e., based on a small number of task-specific labeled examples and requiring a small amount of computation.

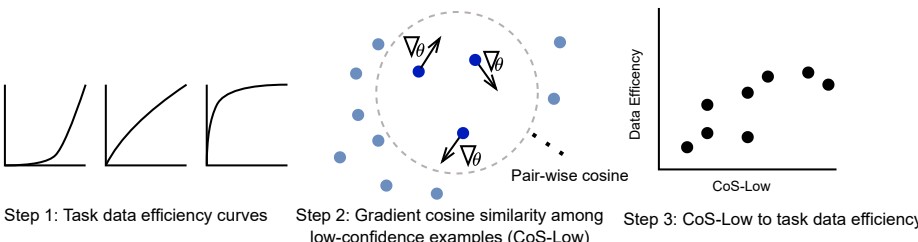

Figure 1: Overview of our approach to predict task specific data efficiency from a few labeled data samples. We use `CoS-Low` to predict task data efficiency measured from 30 downstream tasks.

In this work, we propose a method that meets our desiderata for estimating data efficiency (Fig. 1). Specifically, we first introduce a precise definition of data efficiency based on the area under the data budget/task performance curve. We then explore different cheaply computable notions of task difficulty and ultimately find that the per-sample gradient cosine similarity of low-confidence examples (`CoS-Low`) is highly correlated with our notion of data efficiency, even when computed over a small number of labeled examples. We then formulate a procedure for estimating data efficiency that maps `CoS-Low` to a parametrized approximation of the data efficiency curve. We validate the effectiveness of this procedure on a curated set of 30 realistic specialized tasks (spanning applications in law, medicine, and well-known benchmarks) with varying levels of data efficiency. Our approach only requires collecting a small number of labeled examples and does not require fine-tuning or tracking training dynamics, making it a viable option for practitioners in resource-constrained settings that need to determine the number of examples to annotate to reach desired performance on a downstream task.

## 2 ESTABLISHING THE VARIABILITY OF DATA EFFICIENCY

A core assumption in our work is that the data efficiency — i.e., the relationship between the number of examples used for fine-tuning and performance — varies significantly from task to task. To support this assumption, we first curate a diverse set of 30 tasks from multiple domains, including science, medicine, law, finance, sports, customer inquiries, and natural language understanding. These tasks are sourced from popular datasets from HuggingFace as well as well-known benchmarks such as SuperGLUE (Wang et al., 2020), GLUE (Wang et al., 2019), and BIG-Bench (Srivastava et al., 2023). We mainly consider multi-class text classification or question-answering (QA) to allow consistent use of the exact string match accuracy to measure performance. We limit our selection to tasks with a known estimate of human-level performance and consider this to be the maximum attainable performance for each task. To address cases where human-level performance underestimates this ceiling, we use the higher of the human-level and the maximum performance observed within 5000 fine-tuning data budget. Additionally, we mainly consider tasks with at least 5000 available labeled examples so that we can measure performance up to a relatively high data budget. We report on tasks with fewer than 5000 labeled examples if the maximum performance is reached after fine-tuning on the available data points. The set of prompts we used for fine-tuning and evaluation[1] and further details on our chosen tasks are available in Section A.

We fine-tune the Llama 3.1 8B Instruct model (Grattafiori et al., 2024) on our set of downstream tasks to evaluate performance after fine-tuning on varying data sizes. Measuring the performance on every possible fine-tuning data size between 1 and 5000 would require 5000 fine-tuning runs for each task and therefore be prohibitively expensive. Instead, we fine-tune the model with 50, 100, 200, 500, 1000, 2500, and 5000 randomly selected data points. We use full model fine-tuning instead of parameter-efficient fine-tuning (PEFT) techniques such as LoRA (Hu et al., 2021), as PEFT methods can exhibit a notable performance gap compared to full model fine-tuning (Biderman et al. (2024); Zhang et al. (2024)) and greater sensitivity to the choice of hyperparameters. Our choice of hyperparameter and training settings are listed in Section C

---

[1]redacted

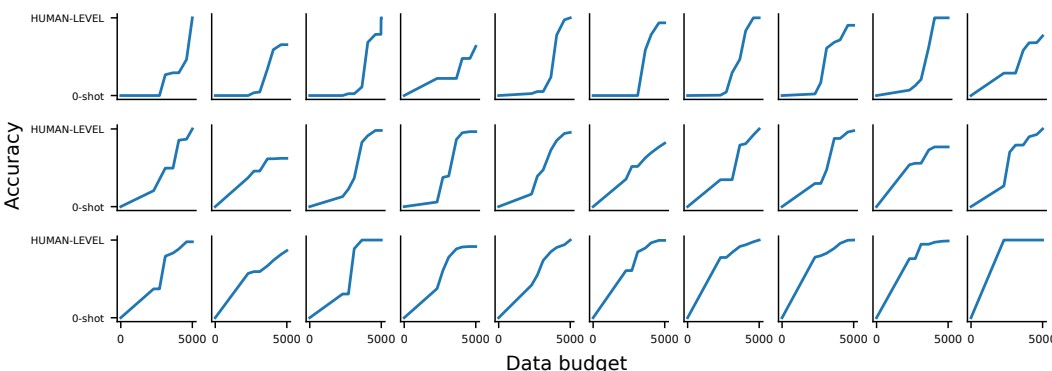

Figure 2: Comparing data budget (from 0 to 5000 examples on log-scale, x-axis) and task performance (from zero-shot to human-level performance, y-axis) across the 30 downstream tasks. The plots are sorted by speed of convergence to the maximum performance level as the fine-tuning data size increases.

Our empirical results, shown in Fig. 2, demonstrate that the scaling relationship between fine-tuning data size and performance is highly task-dependent. Many tasks reach human-expert level performance with fewer than 5000 fine-tuning examples. Some tasks (top rows of figure Fig. 2) show little to no improvement in the lower data regime but display a substantial boost in performance after a certain inflection point. The others (bottom row of Fig. 2) show an almost immediate increase in accuracy with as few as 50 fine-tuning examples. Later in Section 3, we will define a metric that captures this variability in data efficiency.

Interestingly, zero-shot accuracy of a task does not necessarily correlate with task data efficiency; tasks with similarly low or high zero-shot accuracies can have widely different task data efficiencies. We discuss this further in Section A as we report the task-specific raw zero-shot and maximum accuracy.

## 3    MEASURING DATA EFFICIENCY

As discussed above, we informally define the "data efficiency" of a given task as the extent to which we expect performance to improve when annotating and training on additional training examples. In other words, the more data efficient a task is, the fewer data points are required for the model to "solve" it. To better quantify this notion, we require a precise measurement that reflects our definition.

To formulate a reliable metric for data efficiency, we make a series of assumptions inspired by the results of Section 2. First, we assume that, across all tasks, there is limited benefit in annotating additional examples above some maximum data budget. In Fig. 2, performance for many tasks has reached close to human-level performance or otherwise plateaued by 5000 examples, so we assume that for this choice of base model there is limited room for improvement after annotating 5000 examples. This assumption will prove valuable later when we aim to map the prediction of our data efficiency metric back to a concrete estimate of the number of training examples required to reach a certain level of performance. Second, we assume that task performance improves monotonically as the data budget increases. We consider this a safe assumption because, while it is nearly always true in practice, a practitioner would just choose to use a smaller training dataset in cases where training on *more* data results in *worse* performance. In the rare cases where this assumption did not hold in our experiments, we consequently replace the worse performance at the higher budget with the performance at the next-lowest budget, allowing us to capture the gains from additional data and the model's saturation point.

### 3.1    DATA EFFICIENCY DEFINITION

Having motivated our notion of data efficiency and stated our assumptions, we now introduce our proposed metric for concretely measuring data efficiency. Given a fine-tuning data budget $n \in [0, N]$

where $N = 5000$ is the maximum available dataset size, task $k$, and performance function for task $k$ $f_k(n) : n \to \text{acc}_k$ where $\text{acc}_k \in [0, 1]$ is a normalized accuracy that maps the raw zero-shot and maximum attainable (human-level) performance to 0 and 1 respectively, we define area under the curve (AUC) of $f_k(n)$ as the data efficiency measure:

$$\text{AUC}_k = \frac{1}{N} \sum_{n=0}^{N} f_k(n)$$

where $\text{AUC}_k \in [0, 1]$. Mathematically, our notion of data efficiency measures the average performance as a function of the data budget. If $\text{AUC}_k$ is close to 1, this implies that performance saturates early with a small number of labeled examples; if closer to zero, this means little to no improvement is attainable from annotating additional examples.

## 3.2 PREDICTING DATA EFFICIENCY

Knowing the ground-truth performance curve $f_k(n)$ and its $\text{AUC}_k$ for a given task $k$ would inform the optimal fine-tuning data, but these measurements can only be made by fine-tuning the model at varying data budgets, necessitating access to a full fine-tuning dataset as well as sufficient computational resources. However, an accurate estimate of the data efficiency curve could provide answers to valuable questions, such as "how many data points should I collect in order to achieve a desired level of performance?" We therefore turn to devising a method for reliably estimate the data efficiency curve. Notably, such a method is only valuable insofar as it does not require many labeled examples to perform estimation.

As far as we know, predicting the data efficiency of a task using cheap-to-compute metrics has not been explored before. We surveyed existing literature for metrics that can capture different aspect of data efficiency. At a high level, we hypothesize learning difficulty of a task, the *task difficulty*, to be related to its data efficiency, which quantifies how quickly the task is learned given more data. Similar notions of task difficulty have been extensively studied in past work in the context of data taxonomy analysis (Agarwal et al., 2022; Jiang et al., 2021; Siddiqui et al., 2022; Swayamdipta et al., 2020), difficult or mislabeled data identification (Agarwal et al., 2022; Jiang et al., 2021; Li et al., 2024; Pleiss et al., 2020), data selection for efficient training (Mindermann et al., 2022; Paul et al., 2023), model memorization and forgetting (Feldman & Zhang, 2020; Hooker et al., 2021) to name a few, and typically involve a measurement made on the model loss (Mindermann et al., 2022), predictions (Swayamdipta et al., 2020), or gradients (Agarwal et al., 2022; Paul et al., 2023). More concretely, we hypothesize that these measurements might be correlated to our notion of data efficiency. Our approach to predicting data efficiency, therefore, amounts to using task difficulty measurements to predict the task data efficiency $\text{AUC}'_k$ using a simple linear regression (Section 3.3).

**Baselines predictors of task difficulty** As baseline metrics of task difficulty, we consider the *gradient norm* and the *model's confidence*, as they capture different notions of the difficulty of individual data points. Unlike some of the past work tracking the variability of these metrics over the course of training (Agarwal et al., 2022; Paul et al., 2023; Swayamdipta et al., 2020), we simply compute them at inference time with a single step gradient descent on a handful of randomly selected data points. We discuss how each of the baseline predictors are computed in detail in Section B.

The *gradient norm* of the model's weights with respect to the model loss relates to the magnitude of change in parameters required to shift the model's prediction to the target. A larger per-example gradient norm indicates that a larger weight adjustment needs to be made to minimize the model error on the given example. Intuitively, learning a task with high gradient norm examples requires a larger change in the pre-trained model and therefore may require more data. Specifically, we compute per-sample $\text{L}_2$ gradient norm of weights with respect to the cross-entropy loss (Eq. (3)) and aggregate to the task-level (Eq. (4)).

The *model confidence* quantifies the degree of model certainty in its prediction. In the context of pre-training data detection, Shi et al. (2024) demonstrates that high model-assigned probabilities on an input sequence can be used to detect whether the input was seen during pretraining. Extending this idea, we investigate whether high model-assigned probabilities on a model's own prediction serve as a signal of familiarity with the task, possibly indicating higher data efficiency. Alternatively, high model confidence across task examples may indicate that the the typical examples have already been learned. Then fitting on the remaining long-tailed instances must rely on memorization (Feldman &

Zhang, 2020) — which makes learning data inefficient (Achille et al., 2020; Jiang et al., 2021). We compute the model confidence by averaging the model probabilities assigned to the tokens in the predicted target, consisting of the most confidently predicted tokens at each timestep (Eq. (5)). We then aggregate them to the task-level using median (Eq. (6)).

**Our predictor: Gradient cosine similarity** We ultimately find that these preexisting metrics do not serve as sufficiently reliable predictors of data efficiency in our experiments (Section 5). To address this shortcoming, we take inspiration from the multitask learning literature (Liu et al., 2024; Sener & Koltun, 2019; Shi et al., 2023; Yu et al., 2020) that studies the *conflicting gradient* problem, where data points from multiple tasks point in different directions, resulting in suboptimal multitask models. To capture gradient conflict, it is typical to measure the cosine similarity between per-sample gradients of the model's weights with respect to the loss for different examples. Intuitively, the gradient cosine similarity measures the conflict in the learning signal from different task examples. Unlike in the multitask learning methods that aim to minimize gradient conflict, we measure the degree of conflict *within* a single task to estimate the task's learnability. Specifically, we compute the median batch gradient cosine similarities of task examples (Eq. (1)):

$$\texttt{cos\_sim}_k = \text{median}\{\cos(g_i, g_j) \,|\, (x_i, y_i), (x_j, y_j) \in B, i \neq j\} \tag{1}$$

where $(x_i, y_i)$ and $(x_j, y_j)$ are a pair of task data points in $B$, $g_i, g_j$ are the corresponding gradient of the weights with respect to the loss, and $\cos(g_i, g_j)$ measures the cosine similarity of two gradient vectors $\frac{g_i \cdot g_j}{||g_i|| ||g_j||}$. By definition, this metric ignores the magnitude of the gradient updates. In our experiments, we find that $\texttt{cos\_sim}_k$ of the low-confidence segment of task examples (Eq. (2)) is the most predictive of our data efficiency metric:

$$\texttt{CoS-Low} = \text{median}\{\cos(g_i, g_j) \,|\, (x_i, y_i), (x_j, y_j) \in B, i \neq j, B \subseteq U_{0.1}\} \tag{2}$$

Specifically, the batch of examples $B$ is sampled from $U_{0.1}$, the top 10% of the low-confidence task examples. In the active learning literature, some works find that examples with high model uncertainty are the most informative for improving model performance (Dredze & Crammer (2008); Hübotter et al. (2025)). The importance of the low-confidence examples in improving model performance may help explain why our method — which captures the alignment of the learning signals from low-confidence examples — is most effective at predicting data efficiency, defined as how quickly a model improves with additional data. However, further theoretical analysis of our method's effectiveness is left for future work.

### 3.3 MAPPING TASK DIFFICULTY TO DATA EFFICIENCY

Recall that our overarching goal is to find a cheaply computable metric that correlates with our notion of data efficiency (Section 3.1). Given such a correlation, we might hope to be able to predict data efficiency and, consequently, the data budget required to achieve a certain level of performance. Concretely, we use the predicted task data efficiency to estimate the corresponding performance curve $\hat{f}_k(n)$, which informs the number of fine-tuning data size required to reach a target performance. We include a high-level algorithm for efficiently estimating fine-tuning data size to reach a target performance in Section K.

To map one of the aforementioned task difficulty metrics to a task's data efficiency, we fit a simple linear regression to obtain the predicted AUC, denoted by AUC′, as $c * d + I$, where $d$ is one of $\texttt{grad\_norm}_k$, $\texttt{conf\_avg}_k$, $\texttt{cos\_sim}_k$, and $\texttt{CoS-Low}$, and $\{c, I\}$ are regression coefficients. To test each of the metrics on the 30 downstream tasks introduced in Section 2, we use a hold-one-out setting in which all tasks except the held-out task $k$ are used in training to model the regressor AUC′ = $c * d + I$, and AUC′$_k$ is predicted using the fitted line.

We map the estimated task data efficiency, AUC′$_k$, to a specific performance curve $\hat{f}_k(n)$ such that its area under the curve (with both its axes normalized to 0 and 1) is precisely AUC′$_k$, allowing us to concretely predict the fine-tuning data size required for a performance level. However, there are multiple ways to model such a curve defined between 0 and 1 on both axes. Following the stated assumptions on the performance curve $f_k(n)$ (that it is a monotonically increasing curve defined between 0 and 1 on both axes, reaching the maximum performance within the maximum data budget

of 5000), we consider different parametric functions whose area under the curve matches $\text{AUC}'_k$. Among the curves satisfying the assumption, we use the the following power function to model $f_k(n)$:

$$\hat{f}_k(n) = n^p, \text{ where } p = \frac{1 - \text{AUC}'_k}{\text{AUC}'_k}$$

See Section D for details on different parametric curves considered to model a task specific performance curve $\hat{f}_k(n)$ and their fit against the actual curve $f_k(n)$.

## 4 EXPERIMENTAL SETUP

**Metric calculation details.** To justify the use of task difficulty proxies from Section 3 to predict task data efficiency, each metric should incur minimal annotation. Therefore, we require that $\text{grad\_norm}_k$, $\text{cos\_sim}_k$ and CoS-Low only use 32 annotated samples; deriving $\text{conf\_avg}_k$ is completely annotation-free. The per-sample $\text{grad\_norm}_k$ and $\text{conf\_avg}_k$ are aggregated to the task-level using median. Similarly, every pair-wise gradient cosine similarity for computing $\text{cos\_sim}_k$ and CoS-Low is aggregated using median. See Table 8 for detailed computation overhead for each metric calculation. Throughout our experiments, $\text{AUC}_k$ are derived from the data efficiency curve with the data size in log-scale of base 2. $\text{grad\_norm}_k$, $\text{conf\_avg}_k$, $\text{cos\_sim}_k$, and CoS-Low are each used to fit the linear regression line to predict the task AUC for all 30 downstream tasks in a hold-one-out setting (Section 3.3). We repeat the metric calculation on 32 examples end-to-end 10 times to measure the variance in AUC prediction. In addition, to validate the robustness of our results across model families and sizes, we replicate our experiments on Mistral 7B Instruct v0.3 (Jiang et al., 2023) and Qwen 2.5 14B Instruct (Bai et al., 2023) (Section 6).

As CoS-Low requires identifying low confidence examples, we run forward passes on at most 2500 *unlabeled* examples to identify the top 10% lowest confidence examples — i.e., examples with the lowest $\text{conf\_avg}$ — and randomly sample 32 from the segment. In practice, this extra compute cost can be lower, as CoS-Low is robust to noise in the low confidence example selection (Section 6). We run ablation studies calculating $\text{grad\_norm}_k$ and $\text{conf\_avg}_k$ on low confidence examples to fix the sample group as CoS-Low but do not find them to be very effective Section F.2.

**Using LoRA for gradient based metrics.** For calculating metrics requiring model gradients, $\text{grad\_norm}_k$, $\text{cos\_sim}_k$, and CoS-Low, we use rank-64 LoRA adaptors to store the gradients, which is both computation and memory efficient. LoRA gradients hold sufficient information needed to estimate task data efficiency compared to using full model gradients (Section F.3) and avoids the high cost for storing per-sample gradients in memory, which for Llama 3.1 8B Instruct model is $\approx \frac{32}{16} * 8 * 32$ examples $\approx 512\text{GB}$, for a model loaded in half precision.

**Baselines.** We use base_max as an additional baseline that relies on the simple heuristic that there will be no performance increase before fine-tuning on the maximum budget (i.e., 5000 data points). This reflects an implicit assumption when the full annotation budget is used upfront before fine-tuning. As base_max assumes static data efficiency curves across tasks, it serves as an upper bound of data efficiency prediction error.

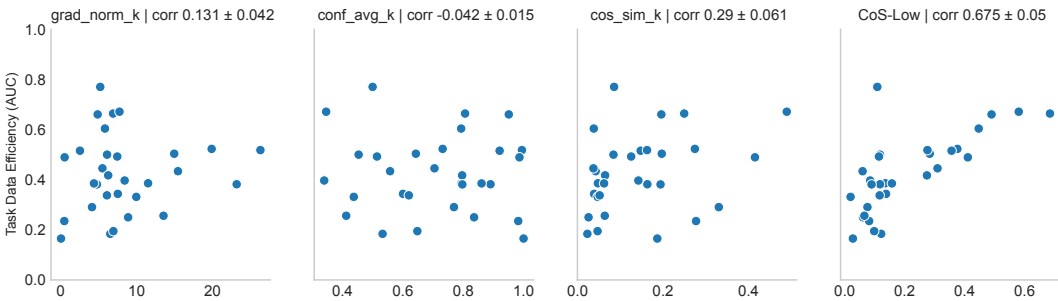

Figure 3: CoS-Low (right) shows the strongest relationship with task data efficiency among other task difficulty metrics. Each metric is compared with the ground-truth task data efficiency (y-axis) using Spearman's rank correlation.

## 5 EXPERIMENTAL RESULTS

Across our experiments, `CoS-Low` shows the strongest performance in efficiently estimating data efficiency and reliably predicts the required fine-tuning data budget across the 30 downstream tasks. To evaluate performance, we report 1) the correlation between each metric and the task data efficiency, 2) absolute mean error of AUC prediction using each metric, and 3) analysis using `CoS-Low` to predict the fine-tuning data size prediction across desired performance levels.

### 5.1 AUC PREDICTION ACCURACY

`CoS-Low` displays the strongest Spearman correlation (0.675) with task data efficiency among all metrics considered (Fig. 3). While past studies track model confidence or gradient magnitude throughout training to surface challenging examples or estimate model's generalization capability (Agarwal et al., 2022; Jiang et al., 2021; Li et al., 2024; Pleiss et al., 2020), our results show that $\text{grad\_norm}_k$ and $\text{conf\_avg}_k$ computed at inference time do not display strong relationships with data efficiency. Consequently, predicting the task data efficiency using linear regression yields a statistically significant result only for `CoS-Low` (p-value $<$ 0.0002) and achieves the lowest prediction error (Table 1).

| Methods | Overall Abs. Mean Error |
|---|---|
| base_max | 0.391 |
| grad_norm$_k$ | 0.130 ± 0.036 |
| conf_avg$_k$ | 0.133 ± 0.036 |
| cos_sim$_k$ | 0.124 ± 0.036 |
| CoS-Low | **0.086 ± 0.030** |

Table 1: Mean absolute error in the AUC prediction using each method. `CoS-Low` (ours) has the lowest overall prediction error (in **bold**) when tested on our downstream tasks for which we measured the actual AUCs.

While `CoS-Low` provides a reliable signal for data efficiency, the relationship is much weaker for `cos_sim_k`, suggesting that gradient similarity provides a stronger signal among low confidence examples than from random examples. However, $\text{grad\_norm}_k$ or $\text{conf\_avg}_k$ computed on the same low confidence segment are not predictive of task data efficiency, which we further discuss in Section F.2. Consequently, we can conclude that `CoS-Low`'s performance stems from the fact that the cosine similarity of gradients is an especially useful signal when computed over low-confidence examples.

### 5.2 FINE-TUNING DATA SIZE PREDICTION ACCURACY

To translate the observed performance of `CoS-Low` into tangible cost savings, we run a cost analysis comparing our task-specific data efficiency prediction method and alternative task-agnostic approaches for finding the optimal fine-tuning data size for the desired performance level. In practice, one can incrementally annotate and repeatedly fine-tune the model until the target performance is reached ("incremental annotation"), or annotate the full dataset (up to 5000 examples) and run a single fine-tuning ("maximum annotation"). `CoS-Low` serves as an in-between approach, where we first fine-tune with the predicted data size, then only train further with incremental annotation approach if the desired performance has not been reached.

We model the fine-tuning cost $C$ as a fixed amount per training run, as the cost of repeated training include access to training resources and human oversight, which does not scale linearly with the dataset size. $A$ denotes the per-example cost of annotation. We assume the number of fine-tuning examples required to reach near-human-level performance is one of 50, 100, 200, 500, 1000, 2500, and 5000. The ground truth data size required to reach the desired performance levels are empirically measured from the 30 downstream tasks.

As shown in Table 2, `CoS-Low`'s approach balance the trade-off between the "maximum annotation" and "incremental annotation" approaches, achieving relatively low excess annotation and few extra training runs compared to either extreme. With the cost of fine-tuning as a function of annotation and compute, practitioners can assess the given annotation and compute cost ratio to adopt a more desirable option. We include further analysis on the fine-tuning data size prediction error of our method in Section E.

| Desired Perf. | Incremental Annotation | | Maximum Annotation | | Ours | |
|---|---|---|---|---|---|---|
| | Extra Annot. | Extra Training | Extra Annot. | Extra Training | Extra Annot. | Extra Training |
| 70% | 0 | $3C$ | $3860A$ | 0 | $219A$ | $1C$ |
| 80% | 0 | $4C$ | $3209A$ | 0 | $748A$ | $1C$ |
| 90% | 0 | $5C$ | $2602A$ | 0 | $701A$ | $1C$ |
| 95% | 0 | $5C$ | $1699A$ | 0 | $1115A$ | $1C$ |

Table 2: Incremental annotation leads to 5 additional fine-tuning runs on average to reach 95% of the human-level performance. Maximum annotation wastes a lot of annotations across all desired performance levels. Even when `CoS-Low` approach underestimates the data size required, necessitating incremental annotation, it only requires one extra fine-tuning run on average and much lower wasted annotation cost.

| Coefficient | Llama 3.1 8B-Instruct | Mistral 7B-Instruct v.03 | Qwen 2.5 14B-Instruct |
|---|---|---|---|
| `CoS-Low` | 0.545±0.005 | 0.797±0.012 | 0.526±0.025 |
| Intercept | 0.310±0.002 | 0.357±0.002 | 0.305±0.003 |

Table 3: Regression coefficient to map `CoS-Low` to data efficiency varies across model families.

## 6 ABLATION STUDIES AND FURTHER ANALYSIS

**Does `CoS-Low` display strong correlation with data efficiency across models families of varying sizes?** To address this question, we repeat our experiments on Mistral 7B Instruct v0.3 and Qwen 2.5 14B Instruct. Training setup and task data efficiency for these model families are discussed in Section G. `CoS-Low` remains the strongest metric for data efficiency prediction across model families (Fig. 12). A key step in our approach is the mapping from `CoS-Low` to data efficiency using the regression weights, and inconsistent regression weights across mode families poses a potential challenge (Table 3). However, we note that the regression weights only need to be computed once for each model and can be reused indefinitely for downstream tasks. Alternatively, the weights can be shared collaboratively within a community to support efficient training.

**Can `CoS-Low` predict data efficiencies of out-of-distribution tasks?** To test the generalizability of `CoS-Low` beyond the original 30 downstream tasks (primarily classification and multiple-choice QA), we extend our experiments to 10 out-of-distribution (OOD) tasks not included in the original set. The OOD tasks comprise two multi-task dataset, four generation tasks, and four domain-specific downstream tasks (see Section H for details). Specifically, we first validate whether `CoS-Low` continues to show strong correlation with task data efficiency among the OOD tasks. We then use their `CoS-Low` and the regression coefficients learned from the original 30 downstream tasks to predict the OOD tasks' data efficiencies.

As shown in Fig. 4, `CoS-Low` reliably predicts how data efficiently each task is learned, with high Spearman's rank correlation of 0.759 (Fig. 4a) and approximately linear trend between `CoS-Low` and task AUC (Fig. 4b). Moreover, once learned, the regression coefficients are reusable to predict data efficiency of unseen tasks. However, the mismatch between accuracy-based performance curves in the original 30 tasks and F1-based curves in generation tasks may introduce errors in the prediction (e.g. the outlier among generation tasks in Fig. 4b), requiring further study.

**When is `CoS-Low` assumption not met?** While empirically observed to be true among the tasks considered, our assumptions that performance reaches a known human-level performance within the given maximum budget may not always be true. For instance, MMLU (multitask accuracy across 57 subjects, spanning various topics from algebraic math to philosophy) (Hendrycks et al., 2021) and MedMCQA (more complex dataset, containing medical entrance exam covering 21 medical subjects and 2,400 healthcare topics) (Pal et al., 2022) are such tasks for which performance remains around $< 75\%$ of human-level with 10,000 training examples.

For such tasks (where the data efficiency curves of these tasks do not follow the proposed $n^p$ curve) we observe that the error in fine-tuning data size prediction grows larger. This failure mode of

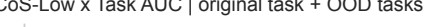

| Task | Corr. with AUC | Abs. mean err |
|---|---|---|
| OOD tasks | $0.727 \pm 0.02$ | $0.109 \pm 0.05$ |
| OOD + Org. tasks | $0.712 \pm 0.04$ | $0.086 \pm 0.03$ |
| Org. tasks | $0.675 \pm 0.05$ | $0.086 \pm 0.03$ |

(a) `CoS-Low`'s correlation with task data efficiency and its task AUC prediction error. `CoS-Low` consistently shows high Spearman's rank correlation with task data efficiency, and the learned coefficients can be reused to predict task data efficiencies.

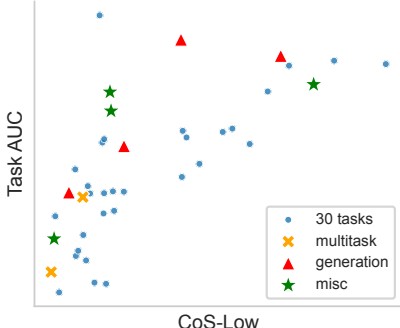

(b) `CoS-Low` and task AUC across 30 original downstream tasks and 10 OOD tasks)

Figure 4: Validating `CoS-Low`'s generalizability in 10 OOD-tasks, consisting of four multitask datasets (orange, cross), four generation tasks (red, triangle), and four domain-specific tasks (green, star).

`CoS-Low` highlights the difficulty of estimating the point of performance saturation for a given task, which may be below the human-level performance. Using human-level performance as a proxy for maximum attainable performance may overestimate the true saturation point of the model, adding noise to the prediction.

**Is the maximum budget of 5000 data points sufficient?** As we use the higher of human-level performance and the maximum observed accuracy within 5,000 examples to approximate the model's maximum attainable performance, it is important to confirm that the performance within this data budget is sufficiently close to the true maximum. To establish this cut-off, we first verify that performance gains beyond 5,000 data budget are marginal across the 30 tasks (Section F.4, Fig. 10a). In addition, we show the impact of increasing the maximum data budget to 10,000 and rerunning the experiments does not add significant changes to our findings (Section F.4, Fig. 10b). This result highlights that `CoS-Low` correlates most strongly with earlier performance improvements at smaller budgets, rather than the small gains observed at the tail end.

**How robust is `CoS-Low` to Sample Size and Low-Confidence Segment?** Throughout our experiments, we select 32 task data samples among the top 10% of low-confidence examples to calculate `CoS-Low`. While we have demonstrated its high correlation with our data efficiency measure, we explore how sensitive our method is to the choice of sample size and low-confidence segment. We vary the sample size and the low-confidence segment and examine 1) the correlation between the newly computed `CoS-Low` and task data efficiency, and 2) the mean absolute AUC prediction error.

We randomly select 4, 8, and 16 examples among the low-confidence segment of the downstream task to compute `CoS-Low` and use them to predict task data efficiency. We find that the relationship between `CoS-Low` and the task AUC becomes weaker (Fig. 8a in Section F.1) and the overall AUC prediction error increases with smaller batch size (Fig. 9a in Section F.1). However, the AUC prediction still has a statistical significance (p-value $<0.05$) using sample size of 8 or 16, suggesting our method is reasonably robust to the choice of sample size.

Another key step in computing `CoS-Low` is the selection of datapoints in the "low confidence segment" of the task dataset. To measure the sensitivity to datapoint selection from the low-confidence segment, we sample task data points from the top 30%, 50%, and 70% of the low-confidence segment. Notably, sampling examples from the top 30% or even 50% of low-confidence segment still produces a Spearman's rank correlation greater than 0.5 with the task data efficiency (Fig. 8b in Section F.1) and results in statistically significant AUC prediction (Fig. 9b in Section F.1). This result indicates that our method can perform well without needing to scan the entire dataset to identify the lowest-confidence examples.

# 7 RELATED WORK

**Data efficiency** In the context of pre-trained LLMs, past work (Aghajanyan et al. (2020); Brown et al. (2020); Sanh et al. (2022); Wei et al. (2022); Zhou et al. (2023)) demonstrates that knowledge is mostly learned during the pretraining phase, allowing for effective knowledge transfer during fine-tuning. However, learning long-tail knowledge requires memorization and typically requires more data (Achille et al. (2020); Feldman & Zhang (2020); Hooker et al. (2021); Jiang et al. (2021)). Zhang et al. (2024) quantifies the impact of fine-tuning data size on the downstream performance to establish a fine-tuning scaling law. For various data efficiency predictors discussed, we take inspiration from multi-task learning and active learning literature. Multi-task learning literature Yu et al. (2020); Liu et al. (2024); Sener & Koltun (2019); Shi et al. (2023); Yu et al. (2020) introduce the concept of *conflicting gradients* among more than two tasks, causing convergence difficulty. Active learning approaches aim to choose which unlabeled training samples should be selected for labeling, using statistics such as model uncertainty (Dredze & Crammer (2008); Hübotter et al. (2025)).

**Task difficulty** Past work that aims to measure task difficulty often examines sample-level statistics tracked over training. Some work tracks the variance of the model confidence (Swayamdipta et al. (2020)) or per-sample gradients (Agarwal et al. (2022)) during training to surface hard or ambiguous examples. Pleiss et al. (2020); Siddiqui et al. (2022) study data taxonomy (e.g. typical, atypical, challenging, mislabeled, etc.) by observing a data point's learning curve during training. Other work aims to select a subset of more challenging or useful examples to learn the task more data-efficiently (Mindermann et al. (2022); Paul et al. (2023)). These works observe that difficult examples tend to be highly ambiguous or without consistent labels, impacting the rate of learning. We refer to these works and use sample-level difficulty proxies to compute task-level difficulty, but our setting differs because we cannot measure training trajectories without performing fine-tuning.

# 8 CONCLUSION

In our work, we introduce a notion of task data efficiency using the AUC of performance curve as the fine-tuning data size increases. We empirically show that data efficiency can vary dramatically across downstream tasks and aim to predict data efficiency by exploring several measures of task difficulty. Our chosen method leverages the median gradient cosine similarity of low-confidence examples, `CoS-Low`, and can efficiently estimate the task data efficiency using as few as 32 task examples. Finally, we show that using our method to find the optimal data size for a desired performance level can save unnecessary annotation or fine-tuning cost incurred when using simple heuristics.

One future direction of our work is to extend our method to generation tasks using non-accuracy based metrics (e.g., BLEU (Papineni et al., 2002) or ROUGE (Lin, 2004) scores, or even LLM-as-a-judge evaluation (Gu et al., 2025)). Another direction is establishing a more rigorous relationship between model evolution after fine-tuning and low-confidence training samples' gradient cosine similarity. Currently, our work focuses on practical implementation with high-level theoretical justification. Lastly, we assume either the highest observed performance within the data budget or human-level performance is the maximum attainable performance on any given model. In future research, metrics derived from model internals, including the ones considered in our work for task difficulty estimation, can be used to check the degree of model saturation and find the model-specific upper-bound.

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

## A  DOWNSTREAM TASK OVERVIEW

We select 30 downstream tasks that span multiple domains including healthcare, law, finance, safety, and other domains requiring natural language reasoning ability. All but three tasks have at least 2500 training examples (*Temporal_sequences* (Srivastava et al., 2023) has 800, *RTE* (Wang et al., 2020) 2241, *Overruling* (Zheng et al., 2021) 1920). Since our data efficiency metric—task AUC—requires evaluating model performance with up to 5000 fine-tuning examples, we extrapolate the performance for these tasks by assuming their peak performance at the maximum available data size is comparable to the performance at fine-tuning data size of 5000.

Table 4 provides a high-level overview of each task, including its zero-shot accuracy, maximum performance after fine-tuning, maximum attainable performance (defined as the greater of known human-level accuracy or the best fine-tuned performance with the 5000-example data budget), and the task data efficiency. We show that neither high or low task zero-shot performance consistently predicts task data efficiency in Fig. 5, highlighting that estimating downstream task data efficiency is a non-trivial problem. Below, we categorize the tasks by their relevant domains and briefly describe each.

### Medical

*Ade_corpus_v2_classification* (Gurulingappa et al., 2012) consists of medical statements indicating the presence of an adverse drug event (ADE=1 or 0), designed to support the extraction of drug-related adverse effects from medical case reports. *MedMCQA* (Pal et al., 2022) is a multiple-choice question dataset derived from a real-world medical entrance exam covering 21 medical subjects and 2,400 healthcare topics.

### Law

Overruling (Zheng et al., 2021) comprises extracted sentences from legal opinions, a subset of which overrule a prior decision (label=1, 0 otherwise).

### Intent Detection

Banking77 (Casanueva et al., 2020) consists of online banking queries labeled with one of 77 pre-defined user intent categories, supporting intent classification in the financial service domain. *Toxic-Chat* (Lin et al., 2023) consists of user prompts collected from the Vicuna online demo, annotated for toxicity in the user prompts. *Circa* (Louis et al., 2020) presents brief question-answer dialogues with ambiguous responses and crowd-sourced ground-truth labels indicating the underlying intention of the ambiguous answer.

### World Knowledge

*CommonsenseQA* (Talmor et al., 2019) evaluates commonsense reasoning ability requiring prior knowledge across a range of target concepts. *MMLU* (Hendrycks et al., 2021) assesses multitask accuracy across 57 subjects, spanning various topics from algebraic math to philosophy. *Sports_understanding* (Srivastava et al., 2023) examines general understanding of sports by presenting plausible or implausible statements related to sports, given specific actions in sports and names of athletes. *Hyperbaton* (Srivastava et al., 2023) tests the ability to identify the correct order of adjectives in given text.

### Logical Deduction and Reasoning

*Boolean_expressions* and *Web_of_lies* (Srivastava et al., 2023) consist of nested boolean logic, presented either in formal notation or natural language, that evaluate to True or False. *Formal_fallacies_syllogisms_negation* (Srivastava et al., 2023) assesses the ability to distinguish between deductively valid and invalid arguments given a premise and corresponding argument. *Object_counting* (Srivastava et al., 2023) evaluates the ability to count simple objects described in a sentence while ignoring irrelevant distractors. *Temporal_sequences* (Srivastava et al., 2023) requires deduction over a sequence of temporally ordered events. *Tracking_shuffled_objects* (Srivastava et al., 2023) tests the ability to track object ownership as the object is transferred among multiple individuals in a sequence of actions.

### Classic Natural Language Inference

*ANLI* (Nie et al., 2020) and *MNLI* (Wang et al., 2019), and *RTE* (Wang et al., 2020) are natural language inference (NLI) benchmarks, each consisting of a premise and a hypothesis, with their relationship categorized as entailment, contradiction, or neutral. *ANLI* is constructed via adversarial human-and-model-in-the-loop procedure; *MNLI* consists of text extracted from speech, fiction, government speech; and RTE comprises news and Wikipedia texts.

**Miscellaneous Natural Language Understanding**

*QQP* (Wang et al., 2019) and *MRPC* (Wang et al., 2019) assess semantic equivalence between pairs of sentences extracted from the Quora discussion forum and online news respectively. *SST-2* (Wang et al., 2019) is a sentiment classification task based on movie reviews. *Fig-QA* (Liu et al., 2022a) evaluates the ability to interpret figurative language given human-written creative metaphors. *WiC* (Wang et al., 2020) is a word sense disambiguation task determining if a polysemous word has the same meaning in two different text snippets.

**Reading Comprehension**

*QuAIL* (Rogers et al., 2020) and *RACE* (Lai et al., 2017) are multiple-choice reading comprehension tasks. *QuAIL* consists of texts extracted from fiction, news articles, blogs, and the Quora forum. *RACE* is based on English exam passages designed for Chinese students aged between 12 and 18; in our experiments, we use the subset containing high-school level passages. *BoolQ* (Wang et al., 2020) consists of a short passage paired with a yes-or-no question related to the passage. *QNLI* (Wang et al., 2019) assesses whether the answer to a question can be inferred from a given paragraph extracted from Wikipedia.

**Visual and Spatial Reasoning**

*MNIST_ascii* (Srivastava et al., 2023) is a multi-label classification task based on the original MNIST dataset, where digits from 0 to 9 are rendered in ASCII string format rather than images. *Reasoning_about_colored_objects* (Srivastava et al., 2023) assesses the ability to understand spatial relationships by interpreting visual descriptions of scenes involving colored objects.

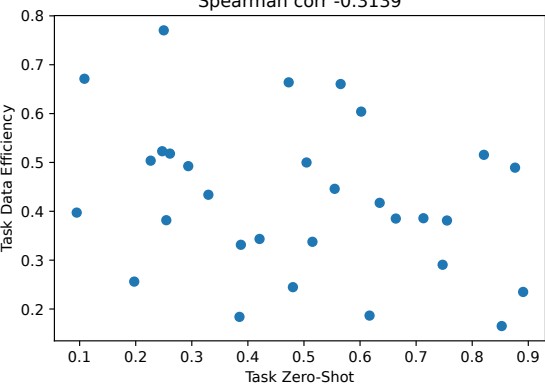

Figure 5: Relationship between zero-shot accuracy and task data efficiency. While higher zero-shot accuracy of tasks close to performance saturation may indicate lower task data accuracy, the relationship is not consistent (Spearman rank correlation of -0.3139).

## B  TASK DIFFICULTY METRIC DEFINITIONS

We compute $\texttt{grad\_norm}_k$ as per-sample $L_2$ gradient norm of weights with respect to the cross-entropy loss (Eq. (3)), aggregated to the task-level:

$$\texttt{grad\_norm}(x_i, y_i) = ||\nabla_w L(x_i, y_i)|| \qquad (3)$$

$$\texttt{grad\_norm}_k = \text{median}\{\texttt{grad\_norm}(x_i, y_i) \mid (x_i, y_i) \in B\} \qquad (4)$$

| Task | Model Accuracy | | Max Attain. Acc. | Task AUC |
|---|---|---|---|---|
| | Zero-shot | Max Fine-tuned | | |
| BoolQ | 0.85 | 0.90 | 0.90 | 0.165 |
| ANLI | 0.39 | 0.74 | 0.92 | 0.184 |
| WiC | 0.62 | 0.86 | 0.86 | 0.186 |
| SST-2 | 0.89 | 0.95 | 0.98 | 0.235 |
| Formal_fallacies_syllogisms_negation | 0.48 | 0.99 | 0.99 | 0.245 |
| Tracking_shuffled_objects | 0.20 | 0.95 | 1.00 | 0.256 |
| MRPC | 0.75 | 0.88 | 0.88 | 0.291 |
| Reasoning_about_colored_objects | 0.39 | 0.94 | 1.00 | 0.332 |
| Web_of_lies | 0.52 | 1.00 | 1.00 | 0.338 |
| MedMCQA | 0.42 | 0.79 | 0.90 | 0.343 |
| QQP | 0.76 | 0.86 | 0.86 | 0.381 |
| MMLU | 0.25 | 0.65 | 0.90 | 0.382 |
| Sports_understanding | 0.66 | 0.99 | 1.00 | 0.386 |
| Boolean_expressions | 0.71 | 0.99 | 1.00 | 0.397 |
| MNIST_ascii | 0.09 | 0.94 | 0.98 | 0.397 |
| MNLI | 0.64 | 0.87 | 0.92 | 0.417 |
| Banking77 | 0.33 | 0.93 | 0.93 | 0.434 |
| Fig_qa | 0.55 | 0.95 | 0.95 | 0.446 |
| Toxicchat0124 | 0.88 | 0.97 | 1.00 | 0.489 |
| QuAIL | 0.29 | 0.84 | 0.84 | 0.492 |
| RACE | 0.50 | 0.84 | 0.85 | 0.500 |
| CommonsenseQA | 0.23 | 0.80 | 0.89 | 0.504 |
| Overruling | 0.82 | 0.97 | 0.97 | 0.516 |
| RTE | 0.26 | 0.88 | 0.94 | 0.518 |
| Object_counting | 0.25 | 0.97 | 0.97 | 0.523 |
| Hyperbaton | 0.60 | 1.00 | 1.00 | 0.604 |
| QNLI | 0.57 | 0.93 | 0.93 | 0.660 |
| Ade_corpus_v2_classification | 0.47 | 0.95 | 0.95 | 0.664 |
| Circa | 0.11 | 0.91 | 0.92 | 0.671 |
| Temporal_sequences | 0.25 | 1.00 | 1.00 | 0.770 |

Table 4: Downstream task's zero-shot accuracy, maximum accuracy after fine-tuning, maximum attainable accuracy (greater of the the human-level performance or the maximum fine-tuned accuracy), and task data efficiency metric (AUC). The tasks are sorted in the order of ascending task AUC, just as in Fig. 2

where $(x_i, y_i)$ is an $i$-th input and corresponding target label with length $T$, from a randomly sampled set of task data points $B$. $L$ is the cross-entropy loss $-\frac{1}{T}\sum_{t=0}^{T}\log P[y_{it}]$, and $P[y_{it}]$ is the probability assigned by the model to the $t$-th target label.

$\texttt{conf\_avg}_k$ is computed by averaging the model probabilities assigned to the predicted target $y_i'$ generated using greedy decoding (Eq. (5)). We then aggregate them to the task-level using median (Eq. (6)).

$$\texttt{conf\_avg}(x_i, y_i) = \frac{1}{T}\sum_{t=1}^{T}P[y_{it}'] \tag{5}$$

$$\texttt{conf\_avg}_k = \text{median}\{\texttt{conf\_avg}(x_i, y_i) \mid (x_i, y_i) \in B\} \tag{6}$$

Note that $T$ is the length of the target label $y$ and is known in advance because our setup mainly considers short generation tasks.

## C    FINE-TUNING SETUP

To measure task data efficiencies, we run full model fine-tuning on Llama 3.1 8B Instruct, Mistral 7B Instruct v0.3 and Qwen 2.5 14B Instruct on each of the 30 downstream tasks (results in Section 2). All experiments are conducted using two Nvidia H100 GPUs for the 8B and 7B models, four for the 14B model, on a high-performance compute cluster. We use a warmup ratio of 0.1, an effective

batch size of 32, a learning rate of 1e-5, and a cosine learning rate scheduler. Models are trained for a maximum of 500 steps, and the reported fine-tuned performance corresponds to the checkpoint with the lowest evaluation loss within the 500 steps. We use early stopping with a patience of 20, terminating training if the evaluation loss does not improve over 20 consecutive logging steps. Training examples with sequence lengths exceeding 2048 tokens are filtered out. All training runs use a fixed random seed for reproducibility.

Rounds of fine-tuning and evaluation with varying fine-tuning data sizes (50, 100, 200, 500, 1000, 2500, 5000) use the same test split within the same task. The test set contains up to 5000 examples. Validation set sizes are capped at 20% of the corresponding training size (e.g., a training set of 50 examples use a validation set of at most 10 examples) to reflect realistic low-resource fine-tuning conditions.

## D  PARAMETRIC CURVE TO MODEL DATA EFFICIENCY

As discussed in Section 3.3, we map the predicted task data efficiency $\text{AUC}'_k$ to a task-specific performance curve $\hat{f}_k(x)$ using a power function $x^p$, where $p = \frac{1-\text{AUC}'_k}{\text{AUC}'_k}$, to model the relationship between fine-tuned performance and fine-tuning data size. In this section, we show an alternative approach using a piecewise linear function Eq. (7) to map $\text{AUC}'_k$ to the performance curve $\hat{f}_k(x)$:

$$\hat{f}_k(x) = \begin{cases} \min\{\frac{1}{2(1-\text{AUC}'_k)} * x, 1\} & \text{AUC}'_k \geq 0.5 \\ \max\{\frac{1}{2\text{AUC}'_k}(x-1)+1, 0\} & \text{AUC}'_k < 0.5 \end{cases} \tag{7}$$

where $x$ is the percentage of the data budget (i.e., data size normalized between 0 and 1). We compare the fit of the predicted performance curves $\hat{f}_k(x)$, estimated using either the power function or the piecewise linear function, with the original performance curves $f_k(x)$ (Fig. 6).

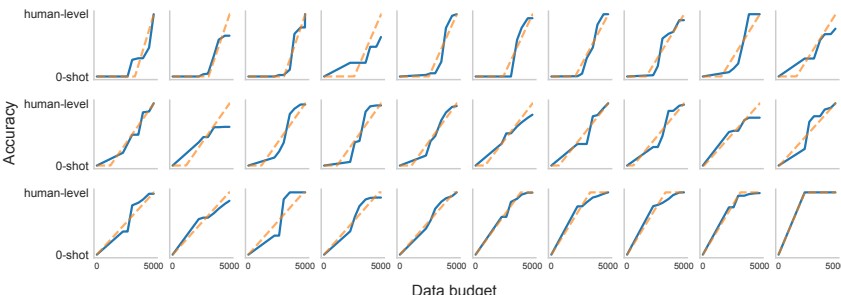

(a) Actual performance curve and the predicted performance curve using linear function.

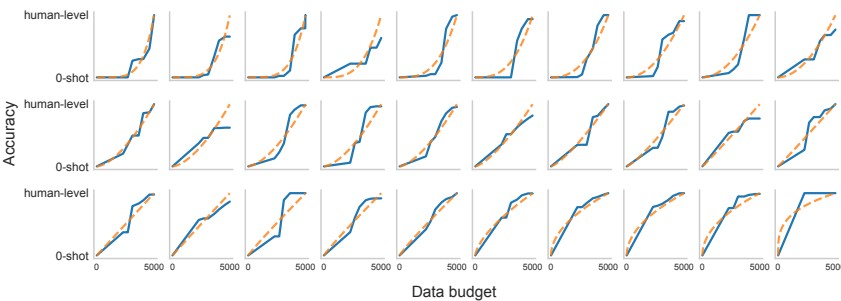

(b) Actual performance curve and the predicted performance curve using power function.

Figure 6: The actual performance curve compared with both power and linear functions.

The absolute error of the fit for the power function is slightly higher, at 8.47% error, whereas the piecewise linear function has 8% error. Despite the marginal difference, we choose the power func-

tion in our analyses as it better captures the gradual performance improvements in the low-data regime, whereas the piecewise linear function introduce a sharp transition.

# E  FINE-TUNING DATA SIZE PREDICTION ERROR

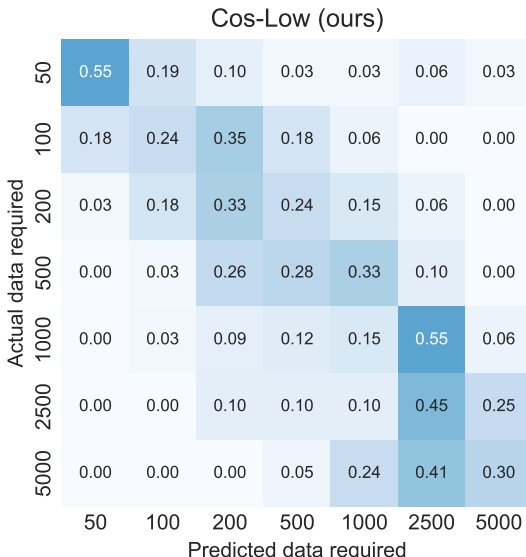

Figure 7: Actual vs. Predicted fine-tuning data size across all desired performance levels between 40% and 95%.

Fig. 7 illustrates CoS-Low's data size prediction error across varying desired performance levels of 40%, 50%, 60%, 70%, 80%, 90%, and 95%. Fig. 7 illustrates that our method is able to identify cases where only a small number of fine-tuning examples are sufficient (illustrated by the darker blue diagonal squares in the upper-left corner of Fig. 7).

# F  ABLATION STUDIES

## F.1  SAMPLE SIZE AND CONFIDENCE SEGMENT

Fig. 8 and Fig. 9 demonstrate that CoS-Low is reasonably robust to both the sample size and the threshold of low-confidence segments. Fig. 8a shows that CoS-Low continues to exhibit a non-random relationship with task data efficiency even when the number of task examples used to compute CoS-Low is less than 32 (our default). In particular, the prediction of task data efficiency made using as few as 8 or 16 examples remains statistically significant (p-value $< 0.05$).

Similarly, CoS-Low derived using confidence thresholds higher than the default top 10% is predictive of the task data efficiency, as reported in Fig. 9b. Although the strength of the relationship becomes weaker, using samples from top 50% low-confidence segment still yields statistically significant meaningful predictions (Fig. 9b).

## F.2  CALCULATING GRAD_NORM_K AND CONF_AVG_K ON LOW CONFIDENCE EXAMPLES

We run additional ablation studies to calculate $\text{grad\_norm}_k$ and $\text{conf\_avg}_k$; gradient norm and average model confidence on the low-confidence examples used for CoS-Low (Table 5). Among all task difficulty metrics considered, CoS-Low (ours) is the most predictive of the data efficiency, which indicates that its predictive power not only comes from the low confidence examples but also from gradient signal conflict from cosine similarity metric.

## F.3  FULL VS. LOW DIMENSION MODEL GRADIENT

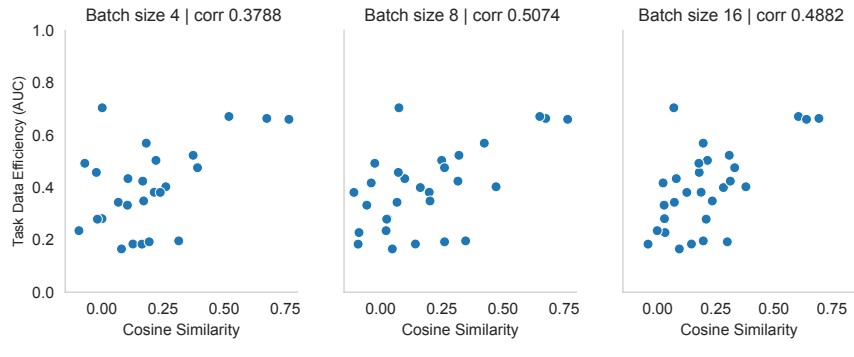

(a) Task data efficiency (AUC) and `CoS-Low`, derived using varying sample sizes of 4, 8, 16.

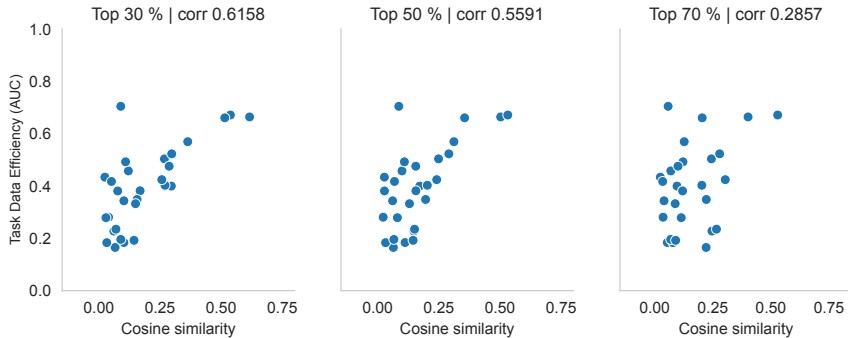

(b) Task data efficiency (AUC) and `CoS-Low`, derived from the top 30, 50, 70% low-confidence segments.

Figure 8: Relationship between the task data efficiency and `CoS-Low`, across varying sample sizes and low-confidence segment. The strength of the relationship is captured using Spearman's rank correlation.

| Batch Size | Mean Abs. Error | p-val |
|---|---|---|
| 4 | 0.121 | 0.068 |
| 8 | 0.111 | 0.015 |
| 16 | 0.103 | 0.0048 |

(a) AUC prediction error by batch size.

| Conf. segment | Mean Abs. Error | p-val |
|---|---|---|
| 30% | 0.095 | 0.00066 |
| 50% | 0.107 | 0.0064 |
| 70% | 0.128 | 0.18 |

(b) AUC prediction error by low-confidence segment threshold.

Figure 9: The mean absolute AUC prediction error across all downstream tasks using `CoS-Low` with varying batch sizes and low-confidence segment thresholds. Fig. 9a and Fig. 9b display statistically significant predictions (p-value $< 0.05$) can be made with the sample size as small as 8 and low-confidence threshold as high as 50%.

We explore the amount of information retained or lost due to using rank 64 LoRA gradient or dimensionality reduction technique such as Gaussian random projection, instead of full model gradients when computing `CoS-Low`. For Gaussian random projection, we randomly project each layer's full gradient to a lower dimension and concatenate them into a single vector to compute the metric.

| Methods | Correlation with data efficiency |
|---|---|
| $\mathrm{grad\_norm}_k$ | $-0.1054 \pm 0.041$ |
| $\mathrm{conf\_avg}_k$ | $-0.1374 \pm 0.023$ |
| `CoS-Low` | $\mathbf{0.675 \pm 0.056}$ |

Table 5: Correlation between data efficiency and task difficulty metrics computed on low-confidence examples.

We do not observe a clear advantage in using a much lower dimensional gradient (Table 6), supporting that using low-rank gradients is an effective and efficient way of computing data efficiency predictor, `CoS-Low`.

| Methods | Correlation with data efficiency | Size of Gradient Vector |
|---|---|---|
| Full gradient `grad_norm_k` | $0.160 \pm 0.021$ | 8GB |
| Full gradient `CoS-Low` | $0.628 \pm -0.052$ | 8GB |
| Rank 64 `CoS-Low` (our choice) | $0.675 \pm 0.056$ | approx. 160M |
| Random Projection `CoS-Low` | $0.630 \pm 0.053$ | approx. 1.6M |

Table 6: `CoS-Low` computed with rank 64 LoRA gradient outperforms alternate approaches and is relatively memory efficient compared to `CoS-Low` computed with full gradient vectors.

## F.4 Increasing the Maximum Data Budget to 10,000

We measure the average raw accuracy at each fine-tuning data size across the 30 tasks, and use tasks with more than 10,000 available data points (15 out of 30) to rerun the experiments end-to-end with the maximum data budget set to 10,000. Fig. 10 shows the

| Data budget | 50 | 100 | 200 | 500 | 1000 | 2500 | 5000 | 10000 |
|---|---|---|---|---|---|---|---|---|
| Avg. acc. | 0.155 | 0.180 | 0.233 | 0.322 | 0.360 | 0.400 | 0.412 | 0.415 |

(a) Avg. raw accuracy at each data budget across the 30 downstream tasks.

| Max. data budget | Corr. with task data efficiency | Abs. mean error |
|---|---|---|
| 5k budget (30 tasks) | $0.675 \pm 0.05$ | $0.086 \pm 0.030$ |
| 10k budget (15 tasks) | $0.699 \pm 0.10$ | $0.085 \pm 0.031$ |

(b) Spearman's rank correlation between `CoS-Low` and task AUC, and the AUC prediction error when performance curves are measured with 10,000 as maximum data budget, instead of 5,000.

Figure 10: Across the 30 tasks, the accuracy gain is marginal when using more than 5000 fine-tuning data points, and `CoS-Low`'s high correlation with task AUC and its AUC prediction error remain stable when higher data budget is used (Fig. 10b).

## F.5 Performance curves under different random seed

We use a fixed seed (seed=123) when sampling data points to fine-tune for fine-tuning across varying data sizes and use these runs to plot the task performance curves (see Fig. 2). To account for performance variance due to randomness in sampling, we repeat Llama 3.1 8B Instruct fine-tuning with different random seeds for all data budgets (50, 100, 200, 500, 1000, 2500, 5000) on the 30 downstream tasks. We report the median raw accuracy for each of the three random runs, along with the accuracy differences between the original and the new random seed runs (Table 7). The resulting median differences in task AUCs are negligible, with -0.006 (seed 48) and -0.017 (seed 37) relative to the original task AUCs.

## G Generalizing Across Model Families

To assess generalizability across model families, we extend our experiments to the Mistral 7B Instruct v0.3 and Qwen 2.5 14B Instruct. We measure task data efficiency (Fig. 11) and compute corresponding task difficulty metrics to predict data efficiency, using the same compute resources and fine-tuning hyperparameters as in the Llama 3.1 8B Instruct experiments.

As shown in Fig. 12, `CoS-Low` consistently demonstrates the strongest correlation with task data efficiency and outperforms alternative metrics such as `grad_norm_k`, `conf_avg_k`, and `cos_sim_k`.

While these results demonstrate that task data efficiency prediction using `CoS-Low` generalizes beyond the Llama 3.1 8B Instruct model, the relationship between task data efficiency and `CoS-Low` is weaker in comparison. One possible explanation is the larger gap between the fine-tuned Mistral

| | Median raw accuracy | | | Median diff. in raw accuracy | |
|---|---|---|---|---|---|
| Data budget | Seed 123 (org) | Seed 48 | Seed 37 | Seed 123 vs. 48 | Seed 123 vs. 37 |
| 0 | 0.492 | 0.619 | 0.558 | 0.026 | 0.026 |
| 50 | 0.67 | 0.728 | 0.708 | 0.028 | 0.028 |
| 100 | 0.702 | 0.73 | 0.747 | 0.024 | 0.032 |
| 200 | 0.784 | 0.777 | 0.771 | 0.023 | 0.020 |
| 500 | 0.816 | 0.818 | 0.817 | 0.028 | 0.018 |
| 1000 | 0.872 | 0.873 | 0.889 | 0.016 | 0.014 |
| 2500 | 0.923 | 0.916 | 0.918 | 0.014 | 0.011 |
| 5000 | 0.937 | 0.930 | 0.927 | 0.010 | 0.009 |

Table 7: Raw accuracy at each data budget across three fine-tuning runs with different random seeds, and the corresponding accuracy difference between the original run (seed=123) and the two additional runs (seed=48, 37).

and Qwen model performance and human expert-level accuracy for some tasks, due to fine-tuning not improving the performance further from their zero-shot performance.

We hypothesize that the weaker relationship may also be partly attributed to the sensitivity of low-confidence example selection to model-specific tokenization. Our current approach selects low-confidence examples based on the lowest average token probabilities. However, for multi-token outputs, simple averaging does not distinguish between uncertain predictions across all tokens and cases where a single high- or low-confidence token skew the average. The Mistral tokenizer encounters this problem, as it represents multi-digit numbers using multiple tokens. To address this problem, we test perplexity-based confidence estimation to select the low-confidence examples, which provides length-normalized uncertainty estimation. As shown in Fig. 12a, we find that perplexity-based low-confidence example sampling ("CoS-Low (PPL)", correlation = 0.52) achieves higher correlation with data data efficiency compared to probability averaging approach ("CoS-Low", correlation = 0.5). This improvement suggests that CoS-Low can benefit from refined low-confidence estimation, especially for tasks involving longer target outputs.

# H    GENERALIZING TO OUT-OF-DISTRIBUTION TASKS

OOD-tasks used to test the generalizability of CoS-Low beyond the 30 original downstream tasks (Section 6) consist of two multi-task datasets, four generation tasks, and four domain specific tasks. For all OOD-tasks, we use the model's maximum performance within the 5,000 data budget as the maximum attainable performance proxy.

Each of the *two multitask datasets* consist of five tasks sampled from the 30 downtream tasks, one consisting of MNIST_ascii, Boolean_expressions, Object_counting, Sports_understanding, Hyperbaton, and the other of Web_of_lies, Reasoning_about_colored_objects, Temporal_sequences, Tracking_shuffled_objects, Formal_fallacies_syllogisms_negation.

The *four generation tasks* are SQuAD 2.0 (Rajpurkar et al., 2018), Disfl-QA (Gupta et al., 2021), QA WikiData (Srivastava et al., 2023), and CoQA (Reddy et al., 2019). For generation task, model perplexity (PPL) on the generated tokens to identify low-confidence segment instead of average probabilities of the generated tokens (see Section J for metrics considered to estimate model confidence).

The *four domain-specific tasks* are intent-classification dataset (Larson et al., 2019), disaster message categorization (Munro, 2012), twitter sentiment analysis from HuggingFace (zeroshot/twitter-financial-news-sentiment), and MMLU-Pro (Wang et al., 2024). The four downstream tasks are either classification or multiple-choice QnA style tasks, similar to the 30 downstream tasks considered, but do not have a reported human-level performance.

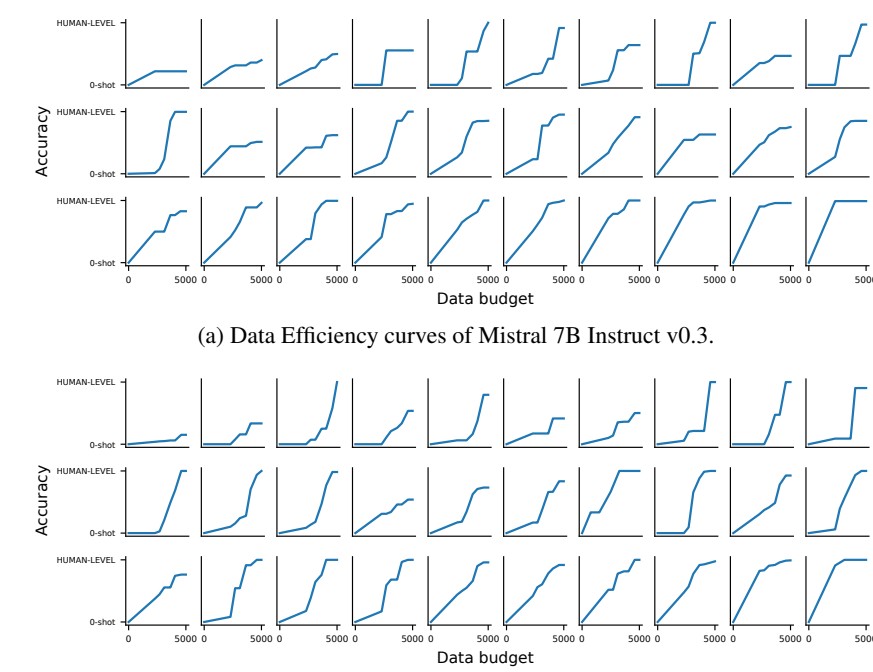

(a) Data Efficiency curves of Mistral 7B Instruct v0.3.

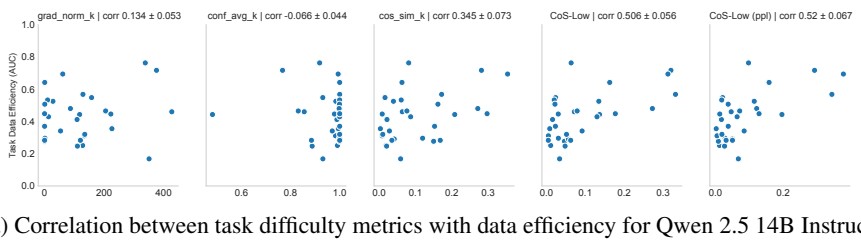

(b) Data Efficiency curves of Qwen 2.5 14B Instruct.

Figure 11: Comparing data budget (from 0 to 5000 examples on log-scale, x-axis) and task performance (from zero-shot to human-level performance, y-axis) across the 30 downstream tasks, for Mistral 7B Instruct v0.3 and Qwen 2.5 14B Instruct.

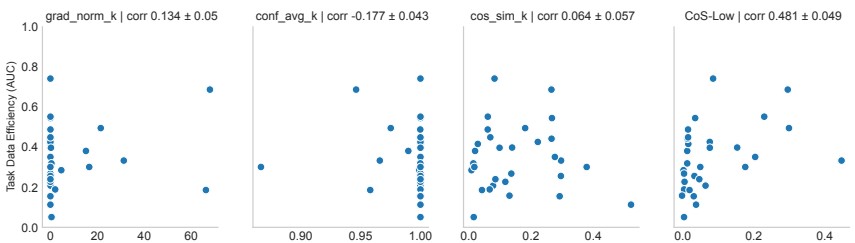

(a) Correlation between task difficulty metrics with data efficiency for Qwen 2.5 14B Instruct.

(b) Correlation between tsak difficulty metrics with data efficiency for Mistral 7B Instruct v0.3

Figure 12: `CoS-Low` shows the strongest correlation with task data efficiency. The relationship is the strongest, however, for Llama 3.1 8B Instruct, followed by Mistral 7B Instruct v0.3 and Qwen 2.5 14B Instruct.

# I COMPUTATION AND MEMORY REQUIREMENT FOR TASK DIFFICULTY METRIC CALCULATION

We assuming backward pass takes twice as much time as forward pass. The main memory requirement is that the model fits in a GPU to be able to run forward and backward pass. The metrics requiring gradients use per-sample gradients of rank 64 LoRAs, takes 1% of the model weights,

| Method | Compute cost | | Memory requirement |
|--------|--------------|--------------|--------------------|
| | Forward pass | Backward pass | |
| $\texttt{grad\_norm}_k$ | 32 * C | 32 * 2C | O(M) |
| $\texttt{conf\_avg}_k$ | 32 * C | | O(M) |
| $\texttt{cos\_sim}_k$ | 32 * C | 32 * 2C | O(M) |
| $\texttt{CoS-Low}$ | 2500 * C + 32 * C | 32 * 2C | O(M) |

Table 8: Compute and memory requirement of calculating task difficulty metrics. C denotes the time of single forward pass, and M the size of the full model.

and adds only a minor memory requirement. $\texttt{CoS-Low}$ requires additional forward passes to identify low confidence examples, but the same number of backward pass as it only uses 32 annotated examples for actual metric calculation.

For an 8B model, peak GPU memory storing the rank 64 LoRA gradients of the 32 samples is (BFloat16 memory) * (model parameter requiring gradients) * (batch size) $\approx$ (2) * (8B parameters * 0.02) * (32) $\approx$ 10 GB. The model loaded on GPU adds an extra (BFloat16 memory) * (full model parameter) = 2 * 8 $\approx$ 16GB.

## J  ESTIMATING MODEL CONFIDENCE

In our exploration of model confidence estimation, many alternatives were considered, including model perplexity on its own generation (PPL) and variational ratio for original prediction (VRO). Among these, $\texttt{CoS-Low}$ on the highest PPL segment showed the strongest correlation with data efficiency than VRO or average softmax probability (our approach). We choose average softmax-probability as confidence proxy for the ease of implementation and reasonably strong correlation with data efficiency; also, it does not require multiple model response generations and requires the least amount of compute. In our experiments, computing PPL required roughly 2x more forward passes, VRO up to 8x. Nonetheless, PPL may be preferred for tasks involving multi-token outputs, especially as average softmax-probability can be skewed by high or low probability tokens as the generation length increases.

## K  USING $\texttt{CoS-Low}$ TO CONCRETELY ESTIMATE FINE-TUNING DATA SIZE

We describe a high-level algorithm using $\texttt{CoS-Low}$ and the regression weights learned from ground-truth AUCs across fine-tuning tasks (i.e. Table 3) to predict the concrete fine-tuning data size required to reach the target performance.

---

**Algorithm 1** Predicting Fine-tuning Data Requirements from $\texttt{CoS-Low}$

---

**Require:** CoS-Low $\ell$, target performance $y$, coefficients $(c, I)$, max budget $N_{\max}$

1: **Step 1: Predict AUC from $\texttt{CoS-Low}$**
2: Predicted AUC: $\widehat{\text{AUC}} \leftarrow c \cdot \ell + I$

3: **Step 2: Estimate required data size**

4: % from data budget needed: $p \leftarrow y^{\frac{\widehat{\text{AUC}}}{1 - \widehat{\text{AUC}}}}$
5: Estimated fine-tuning data size: $n_{\text{required}} \leftarrow 2^{p \cdot \log(N_{\max})}$
6: **return** $n_{\text{required}}$

---

Note that the log transformation is necessary because the x-axis (fine-tuning data budget) of the performance efficiency curve is on a log-scale (Section 4). Log scale captures the rapidly changing model performance improvements at smaller data sizes.

