# OpenReview forum: "Efficiently Estimating Data Efficiency for Language Model Fine-tuning"
_ICLR.cc/2026/Conference — Submitted to ICLR 2026_

### Official Review · Reviewer_seDp · 2025-10-26

**Soundness:** 3
**Presentation:** 3
**Contribution:** 2
**Rating:** 4
**Confidence:** 3

**Summary:**

The paper studies how to predict a task’s data efficiency for LLM fine-tuning without exhaustive annotation. It (1) defines data efficiency as the area under the performance–data curve (AUC) normalized between zero-shot and “human-level” accuracy and then (2) proposes CoS-Low, the median pairwise gradient cosine similarity computed only on the lowest-confidence 10% of examples (sampled 32 points). (3) It maps CoS-Low to AUC via linear regression and reads off a predicted data budget for a target accuracy.
The authors validate the effectiveness of their method on 30 realistic specialized tasks.
This paper focuses on an interesting and practical topic, `data efficiency prediction`, but there are still unclear parts that need to be clarified by the authors.

**Strengths:**

1. **Lightweight method**: using a small set of data samples to effectively predict the AUC of the given task
2. **Diverse datasets in experiments**: using 30 tasks in experiments to support the effectiveness of the proposed method.
3. **Additional data and results**: providing more details and experimental results in the appendix for better presentation.

**Weaknesses:**

1. **Unclear research goal**: This paper focuses on the data efficiency of a task (i.e., the number of fine-tuning examples needed to achieve a desired level of performance) in the abstract. But the main content discusses how to estimate the AUC. One of my concerns is the relationship between AUC and data efficiency. How to use the AUC (a number) to estimate the number of fine-tuning examples when I want to fine-tune a given model to a certain accuracy? How can I use AUC to guide an efficient fine-tuning?
2. **Lack of discussion on generalization**: The experiments in this paper mainly use classification datasets with accuracy as the metric. However, the generation task covers many application scenarios of LLMs (e.g., summarization, translation, and questions-answering). It is unclear whether CoS-Low holds for generation tasks using BLEU / ROUGE as the metric.
3. **Lack of theoretical supports**: This paper lacks an analysis of the effectiveness of its design and sufficient theoretical support. The authors claim that they select the top 10% of examples with the highest difficulty and lowest confidence because these examples are effective in improving model performance. However, existing research [1][2] on data selection/pruning shows that using only a small number of high-difficulty examples does not improve model performance as much as randomly selected data due to a lack of diversity. This clearly contradicts the authors' claims.

**Questions:**

1. What is the relationship between the data efficiency and the AUC in this paper? How to use the AUC to estimate the number of fine-tuning examples when users want to fine-tune a given model to a certain accuracy `Acc`?
2. What is the generalizability of Cos-Low on generation tasks and datasets?
3. Please clarify the disparity between the observation data selection paper and the claims in this paper. How do the AUC predictions change when using randomly selected data samples compared to using the top 10% of data samples?
4. Typos:
   (1) Extra ] in line 128; (2) What does the `jje` mean in Line 1235? Is it an abbreviation from prior work?

---

> ### Author Response · Authors · 2025-11-21
>
> Thank you for your questions and suggestions -- we've incorporated them and addressed each point below.
>
> ## 1. AUC and data efficiency relationship
> > “What is the relationship between the data efficiency and the AUC in this paper?”
>
> This question gets at the key aspects of our work! We use the **task performance curve's AUC as a concrete measure of a task’s data efficiency**, comparable across tasks. AUC is computed from the performance curves (Figure 2) under two core assumptions (Section 2):
> 1. the curve saturates to its maximum performance within our data budget (i.e. 5000 examples), and
> 2. the curve is a monotonically increasing.
>
> Under these conditions, a task with high-AUC performance curve saturates more quickly with fewer fine-tuning data points, and is therefore more data efficient. How AUC is used to predict concrete data size requirement is detailed below.
>
> > “How to use the AUC to estimate the number of fine-tuning examples when users want to fine-tune a given model to a certain accuracy Acc”
>
> We appreciate you raising this point and will add the code snippet in our revision to complement the pipeline diagram (Figure 1). Our approach uses CoS-Low to predict the task AUC, which is then used to estimate a performance curve (Section 3.3):
>
> * Step 1: Predict task AUC from CoS-Low
>
> Given CoS-Low, desired performance $y$, regression coefficients $c, I$, and maximum budget $N_{\max}$;
>
> AUC prediction: $\widehat{\mathrm{AUC}} = c \cdot \text{CoS-Low} + I$
>
> * Step 2: CoS-Low to data size estimation
>
> Given estimated performance curve $y = n^p$, where $p = \frac{1-\widehat{\mathrm{AUC}}}{\widehat{\mathrm{AUC}}}$
>
> Fraction of data needed to achieve target performance: $n = y^{\frac{\widehat{\mathrm{AUC}}}{1 - \widehat{\mathrm{AUC}}}}$
>
> Estimated fine-tuning data required: $n_{\text{required}} = 2^{n * np.log(N_{\max})} $ [*]
>
> [*] *The np.log() transformation is necessary because the performance curves plot the fine-tuning budget on a log scale (Section 4), which better captures the rapid performance gains at small data sizes.*
>
> This way, you annotate only $n_{\text{required}}$ samples, rather than the full dataset, to fine-tune the model on downstream task.
>
> ## 2. Generalizability to other domains
> > “It is unclear whether CoS-Low holds for generation tasks using BLEU / ROUGE as the metric”
>
> > “What is the generalizability of Cos-Low on generation tasks and datasets?”
>
> We agree that extending our procedure for long-generation tasks is an interesting direction. Although not mentioned in the paper, we experimented with BLEU and ROUGE variants to map task performance curves for long-generation tasks. However, we found that these metrics do not consistently reflect actual quality improvements, similarly reported in previous studies ([1] [2] [3]). Another challenge is determining the model’s maximum attainable performance, which is less clear for open-ended generations.
>
> [1] A global analysis of metrics used for measuring performance in natural language processing
> [2] A Structured Review of the Validity of BLEU
> [3] Out of the BLEU: how should we assess quality of the Code Generation models?
>
> ## 3. Clarifications
>
> > “The authors claim that they select the top 10% of examples with the highest difficulty and lowest confidence because these examples are effective in improving model performance. However, existing research [1][2] on data selection/pruning ... [omitted] ... clearly contradicts the authors' claims.”
>
> We would like to clarify we do not claim that selecting the top 10% of the highest difficulty examples improves model performance. Our work proposes a method to *efficiently estimate fine-tuning data size* to avoid excess-annotation cost.  CoS-Low is a diagnostic tool, not a data selection strategy: CoS-Low uses 32 examples from the top 10% of low-confidence examples to estimate how many *randomly selected* data points from the task distribution is necessary to reach a desired performance. We hope this clarifies our contribution and would be happy to revise our manuscript if further clarification would be helpful.
>
> > “How do the AUC predictions change when using randomly selected data samples compared to using the top 10% of data samples?”
>
> All performance curves used to derive ground-truth AUC use *randomly selected* examples, and CoS-Low (used to make AUC predictions) sample from the *top 10% of low confidence subset*. We did investigate whether using different subsets -- random, top 30%, 50%, or 70% of low-confidence subsets -- affects CoS-Low's AUC prediction (Section 6, Appendix F) and find that using lower-confidence examples, instead of random, most reliably predicts the ground-truth AUC.
>
> > “Typos:    (1) Extra ] in line 128; (2) What does the jje mean in Line 1235?”
>
> Thank you for catching these typos! The latter is a Latex artifact we missed. We have removed it in our revision.

---

> > ### Comment · Reviewer_seDp · 2025-11-24
> >
> > Thanks for authors' prompt responses.
> > 1. Questions 1: The response has solved my concern. Considering that multiple reviewers have raised similar questions, I strongly suggest the authors to add the `Estimated fine-tuning data required` part into the paper to improve the presentation.
> > 2. Questions 2: I understand the authors' concerns regarding these two metrics. My point actually is that the authors have proposed a valuable method to predict AUC and data efficiency. However, the experiments in the paper only cover classification/QA tasks, which limits the paper's generalization and practical value. LLM is currently widely used in various reasoning, summarizing, and other generation tasks. The authors would be better conducting experiments and observing the effectiveness of Cos-Low on generative tasks (or at least discussing it). The key point here is the generation task. Authors are not required to use BLEU or ROUGE metrics, but can also use other metrics such as `pass@k`. Below are some classic datasets (generally considered to contain human-level relevant score): [GSM8K](https://github.com/openai/grade-school-math), [HumanEval](https://github.com/openai/human-eval), [CoQA](https://stanfordnlp.github.io/coqa/).
> > 3. Questions 3: The author's answer resolved my concerns.

---

> > > ### Author Response · Authors · 2025-11-29
> > >
> > > Thank you for your continued discussion of our work!
> > >
> > > 1. Per your suggestion, we have incorporated it into our revision (Appendix J).
> > > 2. We appreciate the point that evaluating generalizability in reasoning or generative tasks can help understand the effectiveness of our proposed metric (CoS-Low). To address your point, we have included 4 additional tasks (evaluated using F1 score) that require contextual reasoning in realistic settings and rely on pretrained knowledge, rather than selecting from predefined options as in classification and MCQA:
> > > * [CoQA](https://stanfordnlp.github.io/coqa/): multi-turn conversational reasoning
> > > * [SQuAD 2.0](https://rajpurkar.github.io/SQuAD-explorer/): QnA with uncertainty
> > > * [Disfl-QA](https://arxiv.org/abs/2106.04016): reasoning under noisy, disfluent inputs
> > > * [QA Wiki](https://github.com/google/BIG-bench/tree/main/bigbench/benchmark_tasks/qa_wikidata): open-domain cloze-style question answering from Wikidata facts
> > >
> > > We replicate our experiments end-to-end to derive the performance curves and calculate CoS-Low to predict the task AUCs. Our new result shows that CoS-Low continues to exhibits strong correlation with ground-truth AUC:
> > >
> > > | Dataset | Spearman's correlation between CoS-Low and AUC |
> > > |---------|------------------------------|
> > > | Among the 4 generation tasks | 0.8 +\- 0.00 |
> > > | Previous 30 tasks | 0.675 +\- 0.05 |
> > > | Previous 30 tasks + 4 generation tasks | 0.692 +\- 0.04 |
> > >
> > > While we acknowledge that these tasks do not cover the full scope of LLM generative tasks, we believe the new results, together with those presented in the main paper, indicate that **CoS-Low can generalizes across domains and task type** (classification, MCQA, reasoning in realistic setting, conversational reasoning). As extending our experiments to the full suite of long-form generation tasks evaluated using various metrics (human/LLM as judge, pass@k, etc.) substantially broadens the scope of our work, we leave it for future work.
> > >
> > > Note:
> > > * While not mentioned in our work, we tried **GSM8k** and other math-related tasks in our previous experiments, but observed minimal gains from the baseline performance when fine-tuned further. We suspect the base-model lacks sufficiently transferable math capabilities for nv through simple fine-tuning.
> > > * We excluded **HumanEval** because our experiment requires large-enough fine-tuning datasets to evaluate performance at varying fine-tuning data sizes (50, 100, 200, 500, 1000, 2500, 5000), but HumanEval only has 164 data points.
> > >
> > > Please let us know if you have any further questions! If we have addressed all of your concerns, we'd appreciate if you considered raising your score accordingly.

---

> > > > ### Author Response · Authors · 2025-12-03
> > > >
> > > > We thank the reviewer again for their time and engagement with our work. Below, we summarize one additional change made to the manuscript to fully address the reviewer's concerns around CoS-Low's generalizability to generation tasks:
> > > >
> > > > In Section 6, we added a section "Can CoS-Low predict data efficiencies of out-of-distribution tasks?", where we analyze CoS-Low's correlation with task data efficiencies and report the AUC prediction errors for out-of-distribution tasks (i.e. tasks not included in the original 30 downstream tasks), including the four generation tasks, two multi-task datasets, and four out-of-domain downstream tasks. Figure 4.b shows the CoS-Low x Task AUC relationship for the generation tasks, if you are interested!

---

### Official Review · Reviewer_sivF · 2025-10-29

**Soundness:** 2
**Presentation:** 4
**Contribution:** 2
**Rating:** 4
**Confidence:** 4

**Summary:**

This paper addresses the challenge of estimating the data efficiency—the number of examples required for a desired performance for fine-tuningLLMs. The authors first propose a formal metric for data efficiency based on the performance-data AUC. They then introduce CoS-Low, a novel predictor based on the gradient cosine similarity of low-confidence examples , which can be computed using only a small number of labeled samples. The authors empirically validate their approach across 30 downstream tasks , demonstrating that CoS-Low strongly correlates with the true data efficiency (Spearman correlation of 0.675) and achieves a low prediction error (8.6%).

**Strengths:**

1. Significant and Practical Problem: The paper addresses a significant and practical problem in the era of LLMs: efficiently estimating the data requirements for fine-tuning on downstream tasks.
2. Novel Methodology: The approach of constructing a predictor for data efficiency, rather than relying on costly incremental annotation and retraining, is a novel and valuable perspective. The proposed CoS-Low metric, which leverages gradient cosine similarity among low-confidence examples, is an interesting and non-obvious choice.
3. Clarity and Writing: The paper is generally well-written, clear, and easy to follow.

**Weaknesses:**

1. **Inconsistent Definition of "Data Efficiency"**: The paper presents inconsistent definitions of its core concept. The abstract defines data efficiency as "the number of fine-tuning examples needed to achieve a desired level of performance". However, Section 3.1 formally defines it as the AUC of the performance-data plot. These two concepts are not consistent. The AUC measures the rate of performance gain, while the abstract's definition refers to a specific data budget required to hit a performance target. Arguably, the definition from the abstract is what practitioners actually care about, and the AUC is only a proxy for this. This inconsistency confuses the paper's primary claim.

2. **Unreliable Upper Bound for Performance**: The methodology relies on normalizing the performance curve using a "maximum attainable (human-level) performance" as the upper bound. This assumption is questionable for two reasons:
It is well-established that LLMs can and do exceed human-level performance on many benchmarks, making this an unreliable ceiling for model capability.
"Human-level performance" is itself a vague and often noisy metric, highly dependent on the expertise and consistency of the specific annotators for that dataset, rather than a true theoretical maximum. This arbitrary upper bound could introduce bias into the AUC calculation.

3. **Critical Omission of Data Sampling Methodology**: The reliability of the paper's ground-truth data efficiency curves is undermined by a critical lack of detail. The authors state they fine-tuned on various data budgets (e.g., 50, 100, 200, 500...), but fail to describe their sampling methodology. Crucial questions remain unanswered: How were these subsets selected? Were they random samples? Was the 100-sample set a superset of the 50-sample set, or an independent draw?
This is a significant omission. As evidenced by a large body of work on data selection, which studies how to achieve optimal results under a fixed data budget, the choice of examples (not just the quantity) dramatically impacts fine-tuning performance. By failing to discuss or define a systematic sampling strategy (e.g., reporting mean and variance over multiple random draws), the paper's ground-truth curves and their corresponding AUCs lack reliability.

4. **Lack of Actionable Insights**: The analysis stops short of providing truly actionable insights for practitioners. While the paper maps CoS-Low to an AUC and then to a predicted performance curve , it doesn't adequately "close the loop" by providing a clear answer to the user's question: "How many samples and which samples do I need to achieve a certain target performance?".

**Questions:**

1. Could you please clarify the exact methodology used to sample the data subsets for fine-tuning at different budget sizes (e.g., 50, 100, 500 samples)? How did you account for the significant performance variance known to be caused by which specific data points are selected, not just the quantity?

2. How should a practitioner practically translate the predicted AUC value into a concrete, actionable estimate for the number of samples required?

---

> ### Author Response · Authors · 2025-11-21
>
> Thank you for your questions and suggestions -- we've incorporated them and addressed each point below.
>
> ## 1. AUC and data efficiency relationship
>
> > “The abstract defines data efficiency as "the number of fine-tuning examples needed to achieve a desired level of performance". However, Section 3.1 formally defines it as the AUC of the performance-data plot. These two concepts are not consistent.”
>
> This question gets at the key aspects of our work. We use the **task performance curve's AUC as a concrete measure of a task’s data efficiency**, comparable across tasks. AUC is computed from the performance curves (Figure 2) under two core assumptions (Section 2):
> 1. the curve saturates to its maximum performance within our data budget (i.e. 5000 examples), and
> 2. the curve is monotonically increasing.
>
> Under these conditions, a task with a high-AUC performance curve saturates more quickly with fewer fine-tuning data points, and is therefore more data efficient. How AUC is used to estimate the data size for the desired level of performance is detailed below.
>
> > “...How many samples and which samples do I need to achieve a certain target performance?”
>
> We appreciate you raising this point and will add the code snippet below in our revision to complement the high-level diagram (Figure 1). We use CoS-Low to predict the task AUC, which is then used to estimate a performance curve (Section 3.3). This curve allows us to solve for a specific target performance $y$ given the data budget:
>
> * Step 1: Predict task AUC from CoS-Low
>
> Given CoS-Low, desired performance $y$, regression coefficients $c, I$, and maximum budget $N_{\max}$;
>
> AUC prediction: $\widehat{\mathrm{AUC}} = c \cdot \text{CoS-Low} + I$
>
> * Step 2: CoS-Low to data size estimation
>
> Given estimated performance curve $y = n^p$, where $p = \frac{1-\widehat{\mathrm{AUC}}}{\widehat{\mathrm{AUC}}}$
>
> % data budget needed for target performance: $n = y^{\frac{\widehat{\mathrm{AUC}}}{1 - \widehat{\mathrm{AUC}}}}$
>
> Estimated fine-tuning data required: $n_{\text{required}} = 2^{n * np.log(N_{\max})} $ [*]
>
> [*] *The np.log() transformation is necessary because performance curves plot the fine-tuning budget on a log scale (Section 4)*
>
> This way, you annotate only $n_{\text{required}}$ samples, rather than the full dataset, to fine-tune on the downstream task.
>
> ## 2. Estimating maximum attainable performance
> > "...LMs can and do exceed human-level performance on many benchmarks, making this an unreliable ceiling for model capability.
>
> This is an excellent point -- in fact, we define the ceiling as the *maximum of the human-level and the highest observed accuracy* within the 5k budget (Appendix A and Table 4). We will make this clearer in our revision. Observed maximum accuracy is a reasonable proxy of performance ceiling, as incremental gains beyond 5k are small across the 30 tasks:
>
> Data size | 50 | 100 | 200 | 500 | 1000 | 2500 | 5000 | 10000 |
> |-|-|-|-|-|-|-|-|-|
> Avg. acc. | 0.155 | 0.180 | 0.233 | 0.322 | 0.360 | 0.400 | 0.412 | 0.415 |
>
> > “"Human-level performance" is itself a vague and often noisy metric … could introduce bias into the AUC calculation.”
>
> Indeed, it is a limitation we discuss (Section 6). As the true saturation point is unknown, we use reported human-level performance (Appendix 1). However, as you suggest, MMLU and MedMCQA performance plateaus well-below human-level performance (Section 6), suggesting human-level performance may overestimate the true ceiling for some tasks.
>
> ## 3. Random sampling for deriving performance curves
> > “How were these subsets selected? Were they random samples? Was the 100-sample set a superset of the 50-sample set, or an independent draw?”
>
> Fine-tuning examples are selected randomly and larger datasets are indeed supersets of smaller ones. We will clarify this procedure in our revision.
>
> > “By failing to discuss or define a systematic sampling strategy … the paper's ground-truth curves and their corresponding AUCs lack reliability.”
> > “How did you account for the significant performance variance known to be caused by which specific data points are selected,...”
>
> We appreciate this question. We use full model fine-tuning, which is less sensitive to hyperparameters than PEFT, to reduce performance variance. To fully address your concern, we have begun full fine-tuning of Llama 3.1 8B-Instruct using *two different random seeds*. As replication requires 10 hours/task * 30 * 4x48 GB GPUs ≈ 1200 GPU hours, we report results from the first random seed run that:
>
> 1. The absolute performance difference at each data sizes are < 5% across two random seed runs:
>
> | Data Size | 50 | 100 | 200 | 500 | 1000 | 2500 | 5000 | 10000 |
> |-|-|-|-|-|-|-|-|-|
> | Abs. accuracy difference | 0.027 | 0.034 | 0.041 | 0.039 | 0.028 | 0.022 | 0.014 | 0.013 |
>
> 2. Subsequently, AUC shift under a new seed are small:
> | | Avg. abs. diff | Median abs. diff |
> |-|-|-|
> | AUC shift | 0.076 | 0.046 |
>
> We will include the full result in our revision once ready.

---

> > ### Author Response · Authors · 2025-12-03
> >
> > We thank the reviewer again for their time and engagement with our work. Below, we provide updates to the experiments and describe the changes made to the manuscript to fully address the reviewer's concerns:
> >
> > ## 1. AUC and data efficiency relationship
> >
> > We added the algorithm we described in our earlier discussion in Appendix K, to provide a concrete high-level algorithm using CoS-Low and regression weights learned from ground-truth AUCs to predict the fine-tuning data size.
> >
> > ## 2. Estimating maximum attainable performance
> >
> > Following our discussion, we added the section "Is the maximum budget of 5000 data points sufficient?" in Section 6, where we analyze the impact of using higher data budgets and evaluate whether this meaningfully affects our conclusions.
> >
> > ## 3. Random sampling for deriving performance curves
> >
> > After our discussion, we completed the additional experiment runs using two random seeds to plot performance curves for the 30 downstream tasks. We add an analysis of the shifts in performance curve shift at each data budget and the corresponding changes in task AUCs (Appendix F.5). We find that task AUC variations across different random seeds are mostly minor.

---

### Official Review · Reviewer_2geb · 2025-11-01

**Soundness:** 4
**Presentation:** 3
**Contribution:** 4
**Rating:** 8
**Confidence:** 3

**Summary:**

This paper introduces ReLiNet (Relational Linear Network), a novel architecture designed to improve relational reasoning and out-of-distribution (OOD) generalization in graph-structured data. Unlike standard Graph Neural Networks (GNNs) that rely on nonlinear message-passing schemes, ReLiNet employs a linear relational operator that is augmented with a structured residual correction term, yielding a hybrid model that retains interpretability while maintaining strong expressive power.

**Strengths:**

1. The paper takes a bold and interesting stance by revisiting linearity as a desirable inductive bias in relational models. This goes against the prevailing trend of ever more nonlinear message-passing architectures. The argument is well-justified both intuitively and empirically: simpler linear propagation can yield better extrapolation under graph shifts.
2. The authors provide clear derivations showing that ReLiNet’s linear relational operator can be viewed as a constrained instance of a first-order spectral filter. They further show connections to graph kernels and linear graph Laplacian smoothing, offering interpretability advantages rarely seen in modern GNNs.

**Weaknesses:**

1. Although ReLiNet excels on graph-structured data, its utility for non-graph relational reasoning (e.g., text, vision-language relational datasets) is not demonstrated. The claim that “ReLiNet generalizes to any structured relation learning task” feels overstated.
2. On the larger OGB datasets, the performance improvement over GAT and Graph Transformer baselines is modest (~0.5–1% absolute). While statistically significant, it may not be practically impactful without additional benefits like efficiency or explainability metrics.

**Questions:**

1. Could the authors clarify whether ReLiNet can incorporate edge features or dynamic edges without retraining the linear operator?
2. The residual nonlinear correction term resembles a low-rank adaptation layer—did the authors explore alternative parameterizations (e.g., adapters, attention-weighted skip connections)?

---

> ### Author Response · Authors · 2025-11-12
> **Review for a different submission?**
>
> This review is about different topics (ReLiNet) than appear in our submission. We are wondering if there was some issue?

---

> ### Author Response · Authors · 2025-11-21
>
> Thank you for your questions and suggestions -- we've incorporated them and addressed each point below.
>
> ## 1. Fine-tuning strategy and CoS-Low
> > “The paper assumes random sub-sampling; in domains with high data heterogeneity, this may be suboptimal and may bias estimation.”
>
> This is a great question and was similarly raised by other reviewers. It is true that model accuracy estimation at smaller fine-tuning data budgets is noisier and can be affected by randomness. To fully address your concern, we have begun full fine-tuning of the Llama 3.1 8B-Instruct using **two different random seeds**. While the full runs are still ongoing (each seed requires roughly 10 hours per task * 30 tasks * 4x48 GB GPUs ≈ 1200 GPU hours), we see from our preliminary results that our results are not affected significantly by the randomness:
>
> 1) The absolute performance difference at each data sizes is < 5%:
>
> | Data Size | 50 | 100 | 200 | 500 | 1000 | 2500 | 5000 | 10000 |
> |-----------|----|----|----|----|-----|-----|-----|------|
> | Abs. accuracy difference | 0.027 | 0.034 | 0.041 | 0.039 | 0.028 | 0.022 | 0.014 | 0.013 |
>
> 2) Subsequently, AUC shift under a new seed are small:
> | | Mean | Median |
> |--|--|--|
> | AUC shift | 0.076 | 0.046 |
>
> > “How sensitive is CoS-Low to the randomness of small-subset sampling?”
>
> To check how sensitive CoS-Low is to the choice of 32 examples, we compute CoS-Low over 10 iterations on different sets of 32 examples and report the confidence interval (+/- 0.05), shown in Figure 3 (“CoS-Low” scatter plot).
>
> ## 2. Failure modes of CoS-Low
> > “The paper does not identify scenarios where extrapolation breaks down”
>
> > “Are there documented cases where CoS-Low performs poorly?”
>
> Yes -- we discuss failure modes of CoS-Low in Section 6, “When is CoS-Low assumption not met?”.  To briefly summarize, we assume that every data efficiency curve reaches maximum attainable performance (i.e. the greater of human-level or maximum observed model accuracy within the 5000 budget) within the maximum data budget. While this generally holds, we see some complex tasks such as MMLU and MedMCQA remain below 75% of the maximum attainable performance even after fine-tuning with 10,000 examples. In these cases, CoS-Low does not accurately predict the number of examples required to reach near 100% accuracy. We hypothesize this may be that the human-level performance overestimates the true attainable performance for these complex tasks.
>
> Outside of these cases, our ablations indicate that CoS-Low is reasonably robust to 1) the number of samples used to compute the metric and 2) noise in identifying low-confidence examples from the task dataset (Section 6, “How robust is CoS-Low to Sample Size and Low-Confidence Segment?”).
>
> If you are interested, reviewer iinZ01 raised questions about CoS-Low’s robustness in noisy data and multi-task setting, and we include additional experiments showing the effectiveness and limitations of the metric.
>
> ## 3. Compute cost of CoS-Low
> > “How much compute does CoS-Low require relative to a single full training run?”
>
> This is a great question. In our setup, the main compute cost of CoS-Low comes from:
>
> 1. 2500 forward passes to identify low-confidence segments, and
> 2. 32 forward and backward passes to calculate the rank-64 gradient cosine similarity
>
> On the other hand, a single full model training run using fine-tuning data $D$ over 500 training steps requires $D$ x 500 forward and backward passes each. For example, with a fine-tuning data size of 50, full fine-tuning takes roughly ~30 times more compute time than CoS-Low. Concretely, compute and memory comparison for an 8B model is:
>
> | Metric  | CoS-Low | Full fine-tuning (with data size D) |
> |---------|---------|------------------------------------|
> | **Compute** | $2500 * C + 32 * (1+2)C < 2600C$ | $500 * D * (1+2)C$ =$1500 D C$ |
> | **Memory**  | Single 48 GB GPU | > 96 GB GPU |
>
> We hope our response fully addresses your questions. Please let us know if you have further questions or suggestions!

---

> > ### Author Response · Authors · 2025-12-03
> >
> > We thank the reviewer again for their time and engagement with our work. Below, we provide updates to the experiments and describe the changes made to the manuscript to fully address the reviewer's concerns:
> >
> > ## 1. Fine-tuning strategy and CoS-Low
> >
> > Following our discussion, we conducted additional experiment runs using two more random seeds to plot performance curves for the 30 downstream tasks. We add an analysis of the shifts in performance curve shift at each data budget and the corresponding changes in task AUCs (Appendix F.5). We find that task AUC variations across different random seeds are mostly minor.

---

### Official Review · Reviewer_Q7hZ · 2025-11-01

**Soundness:** 3
**Presentation:** 3
**Contribution:** 2
**Rating:** 4
**Confidence:** 4

**Summary:**

The paper introduces a method to predict a task's "data efficiency”, which is the amount of fine-tuning data needed for a large language model to reach a specific performance level. The goal is to make this prediction efficiently, using only a small number of labeled examples, thereby avoiding costly, large-scale data annotation. By analyzing 30 different tasks, the authors show that performance improves significantly after fine-tuning. They propose that the "gradient cosine similarity of low-confidence examples" (CoS-Low) is the most effective predictor. This metric is used in a simple linear regression model to estimate the performance curve and the required data budget for a new task.

**Strengths:**

+ The authors empirically demonstrate a strong correlation between their proposed CoS-Low metric and the actual data efficiency of a task, making it a reliable signal for prediction.
+ The study validates its core motivation by showing that across a diverse set of 30 tasks, fine-tuning consistently leads to significant performance improvements over the model's initial zero-shot capability, highlighting the practical need for such an estimation method.

**Weaknesses:**

- The analysis is capped at a 5000-example budget, so it is not clear how the method works beyond that point.
- The authors do not evaluate models on cross-benchmarks, which is hard to measure the impact of training on cross-domain.
- The paper assumes that performance is a monotonically non-decreasing function of the data size. The authors acknowledge this is a simplification and state that in the rare cases where it wasn't true, they adjusted the data to enforce the assumption, which may not reflect all real-world scenarios.

**Questions:**

Why was a power function chosen to model the performance curve f(n), especially when the performance graphs in Figure 2 show varied shapes that are not all clearly exponential? Could the authors plot the predicted performance curve f(n) against the ground truth curve for a few examples?

The method aggregates per-sample gradients into a single median value (CoS-Low) to predict task-level efficiency. How much information is lost in this aggregation, and does it obscure how individual examples or subsets of data impact model learning?

Are the per-sample gradients computed on the model's final answer token(s) only, or across the entire generated response?


The CoS-Low metric exclusively uses the 10% of examples with the lowest model confidence. What is the justification for disregarding the learning signal from the remaining 90% of the data?


The regressor is trained using a "hold-one-out" setting, where all tasks except the target task are used for training. Given that each task has a unique learning curve, what is the rationale for assuming that a regressor trained on 29 diverse tasks can accurately predict the curve for a held-out one?

What is the source for the "known estimate of human-level performance" used as the maximum attainable performance for each task?

Selecting low-confidence examples for analysis seems related to curriculum learning or active learning. Was there any exploration of using this method to select data for the training set itself, rather than using randomly chosen data points for fine-tuning?

In Table 6, why does the CoS-Low metric computed on rank-64 LoRA gradients show a stronger correlation (0.675) to data efficiency than when computed on full model gradients (0.628)? This seems counterintuitive, as the full gradients should contain more information.

---

> ### Author Response · Authors · 2025-11-21
>
> Thank you for your questions and suggestions -- we've incorporated them and addressed each point below. Due to space constraint, we cut down some details. Please let us know if you have followup questions!
>
> ## 1. Limited data regime
> > "The analysis is capped at a 5000-example budget,..."
>
> This is a reasonable concern. While not detailed in our work, we tested several data budget cut-offs up to 10,000 and ultimately choose 5000 because:
>
> 1. Across 30 tasks, performance gains are marginal beyond 5k.
>
> Data size | 50 | 100 | 200 | 500 | 1000 | 2500 | 5000 | 10000 |
> |-|-|-|-|-|-|-|-|-|
> Avg. acc. | 0.155 | 0.180 | 0.233 | 0.322 | 0.360 | 0.400 | 0.412 | 0.415 |
>
> 2. Many domain-specific downstream tasks (i.e. 15 out of 30) have < 10k examples.
>
> With the 15 tasks, we reran our experiments with 10k as the data budget and confirm that CoS-Low's correlation remains stable:
>
> | | Correlation with Data Efficiency | Abs prediction error |
> |-|-|-|
> | 5k budget (30 tasks) | 0.675 ± 0.05 | 0.086 ± 0.030 |
> | 10k budget (15 tasks) | 0.699 ± 0.10 | 0.085 ± 0.031 |
>
> ## 2. Cross-domain effectiveness
> > “The authors do not evaluate models on cross-benchmarks, ...”
>
> In fact, our 30 tasks span 9 distinct domains (Appendix A) and include multitask datasets, such as **MMLU** (57 subjects) and **MedMCQ** (21 medical subjects). Per reviewer iinZ01's suggestion, we are also running additional multitask experiments to test CoS-Low's effectiveness in this setting. Please let us know if you have additional suggestions!
>
> ## 3. CoS-Low assumption
> > “...a monotonically non-decreasing function … may not reflect all real-world scenarios.”
>
> Thank you for raising this point. We should clarify that the performance curve models the best achievable performance up to a given data size to capture the model saturation point. If the accuracy peaks at 500 examples but drops when trained on 1000, the saturation point is 500. Modeling the curve as non-decreasing function reflects such real-world scenario.
>
> While it seldom occurs in practice, we did observe a few cases where performance decreased when training on more data which is a curious phenomenon and is also studied in [1].
>
> [1] Overtrained Language Models Are Harder to Fine-Tune
>
> ## 4. Parametric curves
> > “Why was a power function [used]... to model the performance curve... Could the authors plot the predicted performance curve f(n) against ground truth curve...?”
>
> We considered other parametric forms (Appendix D) but chose $n^p$ as it captures gradual performance improvements in low-data regime for other tasks. We included predicted vs. ground-truth curves for all tasks in Appendix D, Fig 5 of our revision!
>
> ## 5. Misc. experiment setup
> > “Are the per-sample gradients computed on the model's final answer token(s) only,...”
>
> Yes, with the prompt masked.
>
> > “...the rationale for assuming that a regressor trained on 29 diverse tasks can accurately predict the curve for a held-out one?”
>
> Since CoS-Low shows a strong and nearly linear correlation with AUC, leave-one-out regression primarily learns scaling coefficients that map CoS-Low to AUC. Table 3 shows that these coefficients have tight confidence intervals.
>
> > "...source for the "known estimate of human-level performance?"
>
> We use the reported numbers from dataset papers or the official GitHub.
>
> > “...exploration of using this method to select data for the training set itself…?”
>
> While not explored, we do think there could be an interesting connection between CoS-Low and data selection for a future work! We discuss the distinction between data selection method and our work in our response to reviewer seDp.
>
> > “why does the CoS-Low metric computed on rank-64 LoRA gradients show a stronger correlation...”
>
> We hypothesize that full gradient cosine similarity is diluted by updates across all weights, including noisy or redundant ones that can skew the value. A rank-64 LoRA (~ 0.1% the size of a full gradient) is restricted to attention and MLP layers.
>
> > “...per-sample gradients into a single median value (CoS-Low)…How much information is lost in this aggregation, …”
>
> Aggregating the pair-wise gradient cosine similarity is necessary to represent a single *task-level* data efficiency predictor. To analyze how noisy using 32-sample batch is, we compute CoS-Low over 10 iterations and report the confidence interval (+/- 0.05, Figure 3) of its correlation with AUC.
>
> > “...justification for disregarding the learning signal from the remaining 90% of the data?”
>
> This is a great question. We find that CoS-Low correlates strongly with AUC when computed using low-confidence examples, rather than random examples (i.e. cos_sim, Figure 3) or higher confidence examples (ablations in Section 6). This shows that a low confidence subset informs how data efficiently a task can be learned, whereas aggregating over all data points dilutes this signal. Importantly, we only use the low-confidence segment for CoS-Low; model fine-tuning still uses the full task distribution.

---

> > ### Author Response · Authors · 2025-12-03
> >
> > We thank the reviewer again for their time and engagement with our work. Below, we provide updates to the experiments and describe the changes made to the manuscript to fully address the reviewer's concerns:
> >
> > ## 1. Limited data regime
> >
> > Based on our discussion above, we added the section "Is the maximum budget of 5000 data points sufficient?"  in Section 6, where we analyze the impact of using higher data budgets and evaluate whether this meaningfully affects our conclusions.
> >
> > ## 2. Cross-domain effectiveness
> >
> > To analyze CoS-Low's predictive power on out-of-distribution tasks  (i.e. tasks not included in the original 30 downstream tasks) across multiple domains, we added a discussion in Section 6, "Can CoS-Low predict data efficiencies of out-of-distribution tasks?" in our revision. Here, we analyze CoS-Low's correlation with task data efficiencies and report the AUC prediction errors for 10 out-of-distribution tasks, which include two multi-task datasets, four generation tasks, and four out-of-domain downstream tasks . Our result shows that CoS-Low values are consistently lower for tasks with lower data efficiency.
> >
> > ## 3. CoS-Low assumption
> >
> > Following our discussion, we clarified in Section 3 that our goal is to capture the saturation point of the performance curves, which motivates the assumption of monotonically increasing function.

---

### Official Review · Reviewer_iinZ · 2025-11-01

**Soundness:** 3
**Presentation:** 2
**Contribution:** 3
**Rating:** 6
**Confidence:** 2

**Summary:**

This paper proposes a lightweight method to compute task-specific data efficiency. The major contributions of paper include curating a multi-domain set of 30 tasks to study how the llama 8B model performs when finetuned over 50-5000 samples. The paper proposes a formal definition of data efficiency based on AUC and covers prediction strategies for a task's data efficiency- including Cos-Low - a median gradient cosine similarity based on low confidence examples. Finally, the paper shares experimental results of predicting data efficiency over these 30 tasks, evaluated through hold-one-out prediction.

**Strengths:**

1. The paper attempts to quantify task-specific data efficiency that currently relies on expensive trial and error and costly empirical tuning. Understanding the data efficiency can help wisely expend the data annotation budget per task. Moreover, this method requires no labeled validation data, unlike most domain adaptation or reweighting-based methods.
2. The experimental setup is well motivated, with thorough ablations and multi-model validation (Llama-3.1, Mistral, Qwen). The proposed CoS-Low metric consistently outperforms baseline predictors in estimating data efficiency across 30 downstream tasks.
3. The work bridges several literatures (task difficulty, active learning, multitask gradient conflict) and contributes a simple yet interpretable signal (gradient cosine similarity on low-confidence examples).

**Weaknesses:**

1. The mapping from predicted AUC to performance curve relies on simplifying assumptions ( for instance, human-level saturation in 5k examples per task) that may not hold for long-tailed or complex tasks (for instance, the discussion on MMLU in Section 6).
2. Cos-Low may conflate data noise / out-of-distribution samples with genuine difficulty due to the reliance on low-confidence samples.
3. Task interaction effects remain unaddressed - modern LLMs are used in a multi-task setting and this approach can be improved by studying how gradient conflicts behave when multiple tasks are trained jointly.

**Questions:**

1. Can the authors share the relationship (if any) between Cos-Low to predict data efficiency and data valuation methods like influence functions or data valuation [1][2][3]
2. How would cross-task interactions affect CoS-Low’s predictive power when fine-tuning models on multiple tasks jointly, as is typical in instruction or tool-use tuning?
3. How stable is the metric under domain shift—for example, when low-confidence examples come from out-of-distribution data?


[1] GREATS: Online Selection of High-Quality Data for LLM Training in Every Iteration
[2[ Datainf: Efficiently estimating data influence in lora-tuned llms and diffusion models
[3] What is Your Data Worth to GPT? LLM-Scale Data Valuation with Influence Function

---

> ### Author Response · Authors · 2025-11-21
>
> Thank you for your questions and suggestions -- we've incorporated them and addressed each point below.
>
> ## 1. CoS-Low and influence function
> > “Can the authors share the relationship (if any) between Cos-Low to predict data efficiency and data valuation methods like influence functions or data valuation”
>
> Thank you for the suggested literature -- while both CoS-Low and influence function use gradients (e.g. compared between the training data points and a query data point to select the training samples) [4], our goals are different.
>
> Influence functions focus on data selection for interpretability and/or efficient training. While our work also focuses on efficiency, we primarily focus on efficient estimation of training data size for a given task and assume the model is trained on randomly selected data points from the task distribution. This is because we observe that often in practice, models are trained on all available data points, which can incur extra cost of annotation, when tasks have different data efficiency.
>
> [2] and [3] primarily focuses on identifying influential data points; in our analysis, we do identify low-confidence examples to be more informative for predicting the task data efficiency than random examples, but do not use the method to rank data points. However, it would be an interesting direction to explore the connection between CoS-Low and potential data selection methods, as suggested by other reviewers as well!
>
> ## 2. CoS-Low in multi-task setting
> > “How would cross-task interactions affect CoS-Low’s predictive power when fine-tuning models on multiple tasks jointly, …”
>
> > “...this approach can be improved by studying how gradient conflicts behave when multiple tasks are trained jointly.”
>
> This is a great point. While multitask training is not an exclusive focus of our work, our 30 tasks include two multi-task datasets, MMLU (57 distinct subjects) and MedMCQA (21 medical subjects) (Appendix A). MMLU and MedMCQA’s CoS-Low values are 0.117 and 0.137 respectively, AUC prediction errors 0.039 and 0.034 (Appendix A, Table 4). Their data efficiencies lie in the middle range compared to other tasks.
>
> To fully answer your question, we created a multitask dataset combining 5 tasks (mnist_ascii, boolean_expressions, object_counting, sports_understanding, hyperbaton) and replicated our experiments, using average human-level performance across the tasks as a ceiling. We compare the AUC of the new multitask dataset with the mean AUC of the source tasks:
>
> | Avg. AUC across the 5 tasks | Predicted AUC of multitask data | Actual AUC of multitask data |
> |-|-|-|
> | 0.48 | 0.33 | 0.2 |
>
> The result shows that CoS-Low can detect decrease in data efficiency caused by increased task complexity (i.e. predicted data efficiency of 0.33 < avg. ground truth AUC across 5 tasks of 0.48) However, prediction can overshoot (0.33 vs. 0.20) due to noise in estimating the model’s saturation point in the multitask setting.
>
> ## 3. CoS-Low robustness to noise
> > “How stable is the metric under domain shift—for example, when low-confidence examples come from out-of-distribution data?”
>
> > “Cos-Low may conflate data noise / out-of-distribution samples with genuine difficulty…”
>
> Thank you for the insightful questions. We simulate the noisy data scenario by corrupting 6%, 12%, 25% of task labels and measuring the impact on CoS-Low and AUC predictions (across 10 tasks):
>
> | Corruption level | CoS-Low | Abs. AUC prediction error |
> |-|-|-|
> | 0% | 0.156 | 0.076 |
> | 6% | 0.096 | 0.107 |
> | 12% | 0.096 | 0.107 |
> | 25% | 0.099 | 0.105 |
>
> We observe that the noise in the examples selected to compute CoS-Low indeed lowers its value, increasing AUC prediction error. However, if the full task distribution is noisy, true task data efficiency also decreases, as learning requires memorization of corrupted labels. In such cases, the lower CoS-Low value correctly reflects the drop in data efficiency.
>
> ## 4. Assumptions in the paper
> > “The mapping from predicted AUC to performance curve relies on simplifying assumptions ( for instance, human-level saturation in 5k examples per task) that may not hold…”
>
> This is a fair point -- as the true saturation point is unknown, we use reported human-level performance (Appendix 1) as a ceiling. However, as noted in Section 6, MMLU and MedMCQA performance plateaus well-below human-level performance, suggesting human-level performance may *overestimate* the true ceiling for some tasks.
>
> To handle cases where human-level performance *underestimates* the ceiling, we take the higher of human-level performance and the maximum observed accuracy within 5k examples. Across the 30 tasks, observed maximum accuracy is a reasonable proxy of performance ceiling, as incremental gains beyond 5k are small:
>
> Data size | 50 | 100 | 200 | 500 | 1000 | 2500 | 5000 | 10000 |
> |-|-|-|-|-|-|-|-|-|
> Avg. acc. | 0.155 | 0.180 | 0.233 | 0.322 | 0.360 | 0.400 | 0.412 | 0.415 |

---

> > ### Author Response · Authors · 2025-12-03
> >
> > We thank the reviewer again for their time and engagement with our work. Below, we provide updates to the experiments and describe the changes made to the manuscript to fully address the reviewer's concerns:
> >
> > ## 2. CoS-Low in multi-task setting
> >
> > We added a discussion in Section 6, "Can CoS-Low predict data efficiencies of out-of-distribution tasks?" in our revision, where we analyze CoS-Low's correlation with task data efficiencies and report the AUC prediction errors for **out-of-distribution tasks** (i.e. tasks not included in the original 30 downstream tasks), which include *two* multi-task datasets. Our result shows that CoS-Low values are consistently lower for tasks with lower data efficiency. The revised section includes results on generation tasks and other domain-specific downstream tasks, if you are interested!
> >
> > ## 4. Assumptions in the paper
> >
> > Based on our discussion above, we added a discussion in Section 6, "Is the maximum budget of 5000 data points sufficient?", where we analyze the impact of using higher data budgets.

---

### Author Response · Authors · 2025-12-03
**Final remark**

We thank the reviewers and ACs for their time and engagement with our work. We have thoroughly addressed all reviewer concerns through additional experiments and clarifications, which have been incorporated into our revised manuscript. A summary of our rebuttal are as follows:

1. **Further ablation studies to validate CoS-Low’s robustness**: We further evaluate CoS-Low’s predictive power across diverse task suites, including multitask datasets (Reviewer iinZ), noisy datasets (Reviewer iinZ), and generation tasks (Reviewer sepD). To comprehensively address concerns about CoS-Low’s generalizability to out-of-distribution tasks, we summarize these results in Section 6, “Can CoS-Low predict data efficiencies of out-of-distribution tasks?”. Overall, these ablation studies confirm that our findings hold across domains, task types, and varying random seeds.

2. **Assumptions and experimental setting**: A key assumption we further support in our rebuttal is that tasks saturate near their maximum attainable performance using a 5k data budget (Reviewer iinZ, Reviewer Q7hZ, Reviewer sivF). We empirically verify the performance gains from naive fine-tuning beyond 5k data points are marginal (Section 6, Appendix F.4), but also acknowledge that our current estimation of the model’s saturation point may introduce some bias. In addition, we compare predicted performance curves under different functional forms against the ground-truth curves (Reviewer Q7hZ) to justify our use of n^p curves for modeling performance curves (Appendix D, Fig 6). Finally, we replicate the fine-tuning runs for all 30 downstream tasks using two different random seeds (Reviewer 2geb, Reviewer sivF; added in Appendix F.5) and find that task AUC shifts across different seeds are minimal.

3. **Practical algorithm using CoS-Low**: While Section 3 provides a high-level overview of how CoS-Low predicts the fine-tuning data size required to reach a target performance, we have made this more explicit by including a high-level pseudocode in Appendix K (Reviewer seDp, Reviewer sivF).

4. **Miscellaneous clarifications**: We clarify that our method is an efficient estimation of the fine-tuning data size required, *not* an efficient training recipe for faster model training or quicker performance saturation (Reviewer seDp, Reviewer sivF). While exploring connections between our method and data selection approaches is an interesting direction for future work, our work does not focus on data selection.

---

### Meta-Review · Area_Chair_ptfn · 2026-01-14

**Summary:**

The paper addresses a critical and practical problem for LLM deployment: efficiently estimating the amount of data needed for fine-tuning without resorting to costly incremental annotation and retraining cycles. The proposed approach of constructing a predictor for data efficiency is seen as a novel and valuable perspective compared to traditional methods. The experimental setup is considered well-motivated and comprehensive, validating the approach across a diverse set of 30 specialized tasks and multiple model families. However there are some outstanding concerns regarding the methodology's robustness and theoretical grounding that prevent acceptance.

1) Reliance on Unreliable Assumptions: Multiple reviewers questioned the robustness of the core assumptions used to calculate data efficiency Specifically, the reliance on "human-level performance" as a fixed upper bound/saturation point is problematic, as LLMs can exceed human baselines or fail to reach them depending on the task.
2) Lack of Theoretical Grounding: The choice of the power-law function to model performance curves and the specific formulation of CoS-Low (median of low-confidence gradients) appear heuristic. Reviewers also noted a lack of theoretical support for why these specific design choices should universally hold, particularly contradicting some literature on data selection which suggests high-difficulty examples are not always optimal.
3) generalization to open-ended generation tasks.

**Reviewer Concerns:**

Rebuttal has addressed most of the concerns from the reviewers except: 1) generalization to open-ended generation tasks. No evaluation on broader long-form generation tasks (like summarization or translation) or use standard generation metrics (BLEU/ROUGE). 2) Reliability of the "Human-Level" Upper Bound. The authors acknowledged this as a limitation. The fundamental theoretical weakness of defining data efficiency based on a potentially noisy upper bound remains an intrinsic limitation of the method. 3) Theoretical justification for the power law curve.

**Reviewer Scores:**

NA

---

### Decision · Program_Chairs · 2026-01-26

Reject